

# Evaluating and ranking Southeast Asia's exposure to explosive volcanic hazards

Susanna F. Jenkins [1], Sébastien Biass [1], George T. Williams [1], Josh L. Hayes [1], Eleanor Tennant [1],

Qingyuan Yang [1], Vanesa Burgos [1], Elinor S. Meredith [1], Geoffrey A. Lerner [1], Magfira Syarifuddin [1],

Andrea Verolino [1]

[1] Earth Observatory of Singapore, Asian School of the Environment, Nanyang Technological University, Singapore, 639754

*Correspondence to*: Susanna F. Jenkins (susanna.jenkins@ntu.edu.sg) and Sébastien Biass (sbiasse@ntu.edu.sg)

**Abstract**

Regional assessments provide a large-scale comparable vision of the threat posed by multiple sources and are useful for prioritising risk-mitigation actions. There is a need for such assessments from international, regional and national agencies, industries and governments to prioritise where further study and support could be focussed. Most existing regional studies

on the threat posed by volcanic activity have relied on concentric radii as proxies for hazard footprints and have focused only on population exposure, often using indices to make first-order estimates of exposure. However, this approach is an oversimplification of volcanic hazards and their associated impacts. We have developed and applied a new approach that quantifies and ranks exposure to multiple volcanic hazards for 40 high-threat volcanoes in Southeast Asia. For each of our 40 volcanoes, hazard spatial extent, and intensity where appropriate, was probabilistically modelled for four volcanic

hazards across three eruption scenarios, giving 697,080 individual hazard footprints plus 19,560 probabilistic hazard outputs. We then developed a GIS framework to overlay the spatial extent of probabilistic hazard footprints with open-access datasets across five exposure categories. Finally, we used our calculated exposure values to rank each of the 40 volcanoes in terms of the threat they pose to surrounding communities. We present *VolcGIS,* an open-source Python code that implements all of the spatial operations required for exposure analysis, available at github.com/vharg/VolcGIS. We

provide all our outputs - more than 6,500 geotif files and 70 independent estimates of exposure to volcanic hazards across 40 volcanoes - in user-friendly format. Results highlight that the island of Java in Indonesia has the highest median exposure to volcanic hazards, with Merapi consistently ranking as the highest threat volcano. Hazard seasonality, as a result of varying wind conditions affecting tephra dispersal, leads to increased exposure values during the peak rainy season





(January, February) in Java, but the peak dry season (January, February, March) in the Philippines. A key aim of our study

was to highlight volcanoes that may have been overlooked, perhaps because they are not frequently or recently active, but that have the potential to affect large numbers of people and assets. It is not intended to replace official hazard and risk information provided by the individual country or volcano organisations. This study and the tools developed provide a road map for future multi-source regional volcanic exposure assessments, with the possibility to extend the assessment to other geographic regions and/or towards impact and loss.

## 1 Introduction

Southeast Asia is one of the most densely populated regions on Earth; it is also home to over 12% (n=173) of the world's Holocene volcanoes and around 15% (n=1,543) of Holocene eruptions (Global Volcanism Program, 2013). Of these Southeast Asian eruptions, 93% (n=1,435) have occurred since 1500 CE, showing the dominance of historical records in reflecting previous eruptive activity. The relatively short timescale of written eruption records

in the region makes capturing the past, and therefore the likely future, range of eruptive activity challenging. There is a need for detailed geological studies to supplement short eruptive records; however, such studies are lacking for many volcanoes around the world because they can be time-consuming, costly and suffer from a lack of deposit exposure, especially in tropical regions such as Southeast Asia (De Maisonneuve and Bergal-Kuvikas, 2020). In addition, the focus in volcanically active areas is often, justifiably, on monitoring and crisis management of

frequently or currently active volcanoes; however, these are not necessarily the volcanoes whose eruptions will affect the most people in the future. For example, the first historical eruption of Galunggung, Indonesia, in 1822 - a Volcanic Explosivity Index (VEI) 5 event - killed >4,000 people after a repose of ~3,000 years (Brown et al., 2017). Where geological studies can be carried out, priority must be given to those volcanoes that pose a major threat to communities, because of the potential magnitude and intensity of the eruption and/or because of the exposure of

communities and their assets to volcanic hazards.

To identify volcanoes that pose a considerable threat to society, previous studies have applied consistent and transferable methodologies to rank multiple volcanoes according to their hazard (e.g. Aspinall et al., 2011; Auker et al., 2015) or their population exposure (e.g. Small and Naumann, 2001; Freire et al., 2019), with some studies combining the two to evaluate 'threat' (e.g. Ewert, 2007; Brown et al., 2015b; Scandone et al., 2016) on a regional

or global scale (Table 1). Such assessments are typically carried out on a volcano-by-volcano basis making it difficult to compare threat across multiple volcanoes and communities.





**Table 1: Previous studies (in chronological order since 2000) that have compared volcanic hazard, exposure and/or a combination of the two ('threat') across multiple individual volcanoes to provide a rank. Hazard or exposure factors are listed when there are three or less factors. Studies that ranked countries or regions, rather than individual volcanoes (e.g. Dilley et al., 2005; Simpson et al., 2011; Freire et al., 2019), and studies that considered the hazard to a site, rather than from a volcano (e.g. Jenkins et al., 2012a,b; 2018), are not included here.**


| Study | Region | Number of volcanoes | Index-based? | Hazard factors | Exposure factors | Highest threat volcano or country |
|---|---|---|---|---|---|---|
| Small and Naumann, 2001 | Global | 1405 | N | None | Population within 10-200 km radii | Gede, Indonesia |
| Ewert, 2007 | USA | 169 | Y | 15 | 10 | Kīlauea, USA |
| Miller, 2011 | New Zealand | 16 | Y | 15 | 10 | Okataina, New Zealand |
| Aspinall et al., 2011 | 16 WB GFDRR priority countries | 439 | Y | 8 | Weighted population counts within 10, 30, 100 km | Not stated |
| Camejo and Robertson, 2013 | Lesser Antilles | 16 | Y | 8 | Weighted population counts within 10 and 30 km | Not stated |
| Auker et al., 2015 | Global | 328 | Y | 6 | None | Not stated |
| Brown et al., 2015a | Global | 1551 | Y | Volcanic hazard index of Auker et al., 2015 | Weighted population count: 10, 30, 100 km | Indonesia |
| Pan et al., 2015 | Global | Not stated | N | Frequency for each VEI, with radii where most deaths occur for PDC, lahar and tephra fall | Population count within VEI defined 'lethal' radii | Indonesia |
| Scandone et al., 2016 | Italy and the Canary Islands | 19 | Y | Time since last eruption and maximum VEI | Population within radii defined by maximum VEI | Campi Flegrei, Italy |
| Retnowati et al., 2018a | Indonesia | 44 | Y | CVGHM hazard maps converted to tephra load | School building repair costs from tephra fall damage | Not stated |
| Ewert et al., 2018 | USA | 161 | Y | 15 | 10 | Kīlauea, USA |
| Nieto-Torres et al., 2021 | Mexico | 13 | Y | 9 | 9 | Tacaná, Mexico-Guatemala |
| **This study** | **Indonesia and the Philippines** | **40** | **N** | **Probabilistic modeling of tephra fall, large clast, dome collapse and column collapse PDC; both short- and long-term assessments** | **Population, buildings, roads, crops, urban areas** | **Merapi, Indonesia (Java in general)** |


Of populations within 10 km of Holocene volcanoes, those in Southeast Asia are the largest and fastest-growing anywhere in the world (Freire et al., 2019). Indonesia and the Philippines alone have been estimated to contain more than 75% of the

global volcanic threat (where threat is a product of an average volcanic hazard index, number of volcanoes and population within 30 km of volcanoes: Brown et al., 2015b). Exposure and threat estimates across multiple volcanoes typically rely on concentric radii around each volcano as a proxy for the spatial distribution of threat to life from volcanic hazards (Table 1). This approach, although facilitating regional and global exposure analyses, overlooks the complexity of hazardous volcanic phenomena and their interactions with external factors (e.g. wind, topography). In a volcanic context, regional

assessments are complicated by the multi-hazard and spatially-varying nature of eruption products, the wide range of hazard and impact mechanisms and the variable knowledge of eruptive records across different volcanic systems. As a result, most existing regional estimates of population exposure to volcanic hazards rely on an overly-simplified hazard footprint extent and intensity. A more robust estimate of exposure to volcanic hazards requires the use of numerical models able to describe the spatial distribution and intensity of volcanic hazards. Identifying reasonable and physically sound eruption source

parameters (ESP) for these models strongly depends on the knowledge of the volcanic system obtained from the geological mapping of past deposits. However, in areas like Southeast Asia, where studies, access and deposit preservation are limited, defining ESPs can be challenging. For this reason, numerical models are often coupled with probabilistic approaches in order to simulate ranges of credible potential future eruptions and environmental conditions, and quantify the likelihood of certain areas being affected by a given hazard. Several regional (multi-volcano) studies have used probabilistic hazard

modelling to quantify hazard (Hoblitt et al., 1987; Hurst and Smith, 2004, 2010; Jenkins et al., 2012a; Biass et al., 2014) and threat (Jenkins et al., 2012b; Scaini et al., 2014), but they have all focussed on tephra hazard and were limited by computing power. As a result, no comprehensive regional, multi-volcano and multi-hazard exposure analysis has yet been achieved, which raises the question as to what extent current global volcanic exposure analyses based on concentric radii around volcanoes are valid.

To address these issues, we developed and applied a new framework to estimate the exposure to volcanic hazards on a volcano-by-volcano basis, with the aim of ranking volcanoes to identify those that pose the greatest threat. The approach couples probabilistically modelled footprints from four volcanic hazards: tephra fall, large clasts, dome collapse and column collapse pyroclastic density currents (PDCs) across three eruption scenarios (representing VEI 3, 4 and 5). The hazard data are then coupled with open-access Geographic Information System (GIS) data to quantify five categories of

exposure (population, buildings, roads, crop and urban areas). A *Python* library named *VolcGIS* was developed to pre-process and perform all geospatial operations required to quantify exposure. We demonstrate the application of our framework on a selection of volcanoes in Southeast Asia that are considered high-threat. To support the differing requirements of volcanic risk management, we consider exposure at two timeframes: i) *short-term*, which assumes an



eruption has taken place, i.e. exposure is conditional upon the eruption scenario: this can provide important values, maps

and assessments in the event of unrest or crisis management; and ii) *long-term*, which accounts for the annual probability

of the eruption scenario occurring: this is valuable for comparing across multiple volcanoes on a like-for-like basis. Both

methods can identify 'hotspots', allowing future, more targeted hazard and risk assessments to be prioritised. Using these

complementary approaches, we ranked the volcanoes in terms of the nature of the volcanic hazard and the type of exposure.

In what follows, we outline our methods, framework and data sources before presenting and discussing our findings

and limitations. The code is published in open source and outputs are provided in Supplementary Material, with the

intention that they be used to further our understanding of exposure to volcanic hazards in this region. The proposed

methodology provides a transferable and evidence-based approach for evaluating volcanic hazard, exposure and threat

across a volcanic region. This study is not intended to replace official hazard and risk information provided by individual

country or volcano organisations (i.e. Indonesia's Centre for Volcanology and Geological Hazard Mitigation, CVGHM,

and the Philippine's Institute of Volcanology and Seismology, PHIVOLCS. Instead, it is designed to address a need from

international, regional and national agencies, industries and governments for large-scale hazard and risk information to

identify and prioritise volcanoes where further study and support should be focussed.

## 1.1 Choosing volcanoes for analysis

Here, we consider Holocene volcanoes from the Smithsonian Institution's GVP (Global Volcanism Program, 2013) located

in Southeast Asia and with at least one recorded VEI 3 or greater eruption (n=48). To further restrict the volcanoes to those

that are more likely to pose a threat to society, we consider the Population Exposure Index (PEI) for each volcano, an index

that accounts for the increased potential for loss of life with proximity to the volcano (Aspinall et al., 2011, Brown et al.

2015b). For our initial subset of 48 volcanoes, we update the PEI values of Brown et al. (2015b) by recalculating population

counts within 10, 30 and 100 km radii using the Landscan 2018, rather than 2011, population dataset and recalculating the

fatality weightings within each radii using the updated fatality database of Brown et al. (2017) rather than Auker et al.

(2013). The revised fatality weightings do not differ much from those of Brown et al. (2015b), remaining at 0.003 within

the 100 km radius, and incurring only small changes at the 30 km (0.03 to 0.07) and 10 km (0.93 to 0.97) radii. We use the

updated PEI to further restrict our 48 volcanoes by considering only those with a PEI of 4 or above, indicating a fatality

weighted exposed population of 10,000 or more (Table 2). Of the remaining 40 volcanoes, 34 are in Indonesia and 6 are in

the Philippines (Figure 1).  Given the relatively large number of volcanoes in Indonesia, and their geographic spread, we

further subdivide the region geographically into (from west to east): Sumatra, Java, Lesser Sunda Islands, Sulawesi,

Halmahera/Banda Sea. The updated PEI remained the same for 20 of our 40 volcanoes, increased for 17 and decreased for

3 (Table 2). The largest change in PEI is +2 for Paluweh volcano in the Lesser Sunda region of Indonesia due to an increase



from ~550,000 to more than 1 million people within 100 km, following the establishment of the new administration regency

of Nagekeo in 2007.

**Table 2: Volcanoes considered for analysis in this study, the exposed and weighted summed population within 100 km (LandScan 2018) and the updated PEI (and change in PEI from that calculated in Brown et al., 2015b). Those volcanoes with a change in PEI are shown in shaded cells. See text for details on how the PEI was updated. Volcanoes are ordered by decreasing weighted**

**summed population <100 km of the volcano. Volcano IDs are used in Figure 1.**

| ID | Volcano | Region | Population (<100 km) | Weighted Summed Population (<100 km) | Updated PEI [Change in rank] |
|---|---|---|---|---|---|
| 1 | Guntur | Java | 24,672,816 | 647,625 | 7 [0] |
| 2 | Merapi | Java | 20,912,606 | 610,759 | 7 [0] |
| 3 | Gede-Pangrango | Java | 41,052,844 | 464,921 | 7 [0] |
| 4 | Cereme | Java | 24,363,615 | 434,438 | 7 [+1] |
| 5 | Galunggung | Java | 23,503,160 | 411,713 | 7 [+1] |
| 6 | Kelud | Java | 21,445,246 | 399,771 | 7 [+1] |
| 7 | Dieng Volcanic Complex | Java | 20,836,400 | 381,995 | 7 [0] |
| 8 | Taal | Philippines | 25,468,937 | 361,479 | 7 [0] |
| 9 | Papandayan | Java | 19,871,707 | 346,022 | 7 [+1] |
| 10 | Mayon | Philippines | 3,800,811 | 307,870 | 7 [+1] |
| 11 | Lokon-Empung | Sulawesi | 1,615,751 | 302,849 | 7 [+1] |
| 12 | Gamalama | Halmahera/Banda Sea | 557,971 | 237,081 | 6 [0] |
| 13 | Lamongan | Java | 13,034,961 | 232,253 | 6 [0] |
| 14 | Tengger Caldera | Java | 19,308,100 | 206,610 | 6 [0] |
| 15 | Agung | Lesser Sunda Islands | 4,932,198 | 173,099 | 6 [0] |
| 16 | Pinatubo | Philippines | 20,263,766 | 143,351 | 6 [0] |
| 17 | Soputan | Sulawesi | 1,672,484 | 135,393 | 6 [+1] |
| 18 | Semeru | Java | 16,809,817 | 121,729 | 6 [+1] |
| 19 | Bulusan | Philippines | 3,070,592 | 119,290 | 6 [+1] |
| 20 | Rinjani | Lesser Sunda Islands | 3,324,266 | 119,250 | 6 [0] |
| 21 | Sinabung | Sumatra | 7,046,711 | 117,027 | 6 [+1] |
| 22 | Ranakah | Lesser Sunda Islands | 939,183 | 103,308 | 6 [+1] |
| 23 | Iya | Lesser Sunda Islands | 851,704 | 97,554 | 5 [0] |
| 24 | Raung | Java | 6,899,109 | 78,405 | 5 [0] |
| 25 | Camiguin | Philippines | 2,216,661 | 63,373 | 5 [0] |
| 26 | Parker | Philippines | 3,493,014 | 61,911 | 5 [+1] |
| 27 | Tangkoko-Duasudara | Sulawesi | 1,332,181 | 39,204 | 5 [0] |
| 28 | Lewotobi | Lesser Sunda Islands | 627,425 | 29,233 | 4 [-1] |
| 29 | Gamkonora | Halmahera/Banda Sea | 702,145 | 26,855 | 4 [0] |
| 30 | Krakatau | Java | 6,376,553 | 26,122 | 4 [0] |
| 31 | Awu | Sulawesi | 74,125 | 25,829 | 4 [0] |
| 32 | Lewotolo | Lesser Sunda Islands | 388,713 | 24,479 | 4 [0] |



| 33 | Karangetang | Sulawesi | 90,664 | 22,496 | **3 [+1]** |
|----|-------------|----------|--------|--------|------------|
| 34 | Leroboleng | Lesser Sunda Islands | 603,314 | 22,339 | **4 [-1]** |
| 35 | Dukono | Halmahera/Banda Sea | 536,125 | 16,989 | **4 [+1]** |
| 36 | Paluweh | Lesser Sunda Islands | 841,119 | 15,544 | **4 [+2]** |
| 37 | Suoh | Sumatra | 1,526,998 | 13,702 | **4 [+1]** |
| 38 | Iliwerung | Lesser Sunda Islands | 388,155 | 13,199 | **4 [+1]** |
| 39 | Tambora | Lesser Sunda Islands | 975,708 | 11,353 | 4 [0] |
| 40 | Banda Api | Halmahera/Banda Sea | 6,588 | 10,829 | **4 [-1]** |

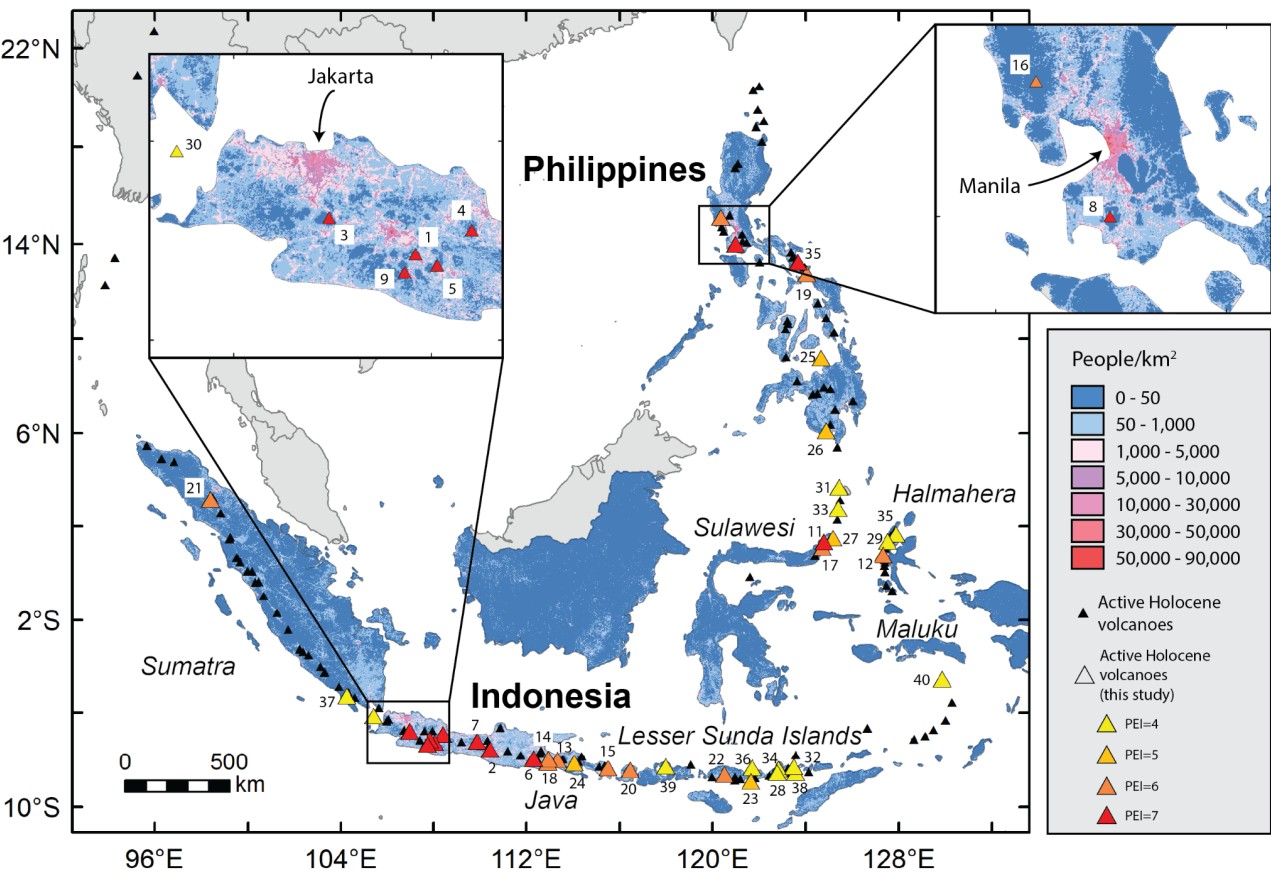

**Figure 1: Active Holocene volcanoes of Southeast Asia (black triangles, as defined in Global Volcanism Program, 2013) with the 40 volcanoes considered for analysis in this study highlighted as larger triangles, with their colour dictated by their PEI. Basemap is ambient population per 1 km² (Landscan 2018). Numbers relate to the volcano name and PEI in Table 2.**





## 2 Methodology

This paper presents a methodology to i) assess the probabilistic hazard associated with short-lived, explosive eruptions of VEI 3, 4 or 5, and ii) estimate various aspects of exposure to these hazards (e.g. population, buildings, roads, urban areas and crops). We considered four hazards produced by explosive volcanic eruptions: i) the static

load caused by tephra accumulation, ii) the kinetic impact associated with large clasts, and inundation from PDC generated from iii) dome collapse and iv) column collapse. A total of 697,080 individual model runs were carried out. For each hazard, the spatial extent, and where appropriate intensity, was modelled for the different eruption scenarios, with results analysed for three differing probabilities: 10%, 50% and 90%, giving a total of 57 permutations of hazard and 285 estimates of exposure per volcano. For tephra fall, further aggregation was carried out per month to identify any seasonal

variability in hazard footprints, producing 324 additional probability aggregated hazard footprints per volcano. Hazard modelling outputs and their associated exposure estimates were coupled with eruption frequency-magnitude estimates to allow two separate rankings to be developed: conditional (assuming the eruption scenario had occurred) and absolute (weighted by the eruption scenario's probability of occurrence). Hazard and exposure were combined using the newly developed *VolcGIS* framework.

### 2.1 GIS framework

We have developed a geospatial python framework that can source multiple derived hazard and exposure datasets of varying resolution, unifying them to one consistent grid (Figure 2). The GIS framework was written in Python 3.9 and relied on Rasterio 1.1.8, rioxarray 0.4, GDAL 3.1.4 and Numpy 1.19.4 for raster analyses and GeoPandas 0.8.1 and Shapely 1.7.1 for vector analyses. For each volcano, the extent of the study area was defined based on the bounds of the 1 kg/m$^2$

tephra isomass occurring at a 10% probability for a VEI 5 eruption (see Section 3.2.1). The GIS first applies preprocessing functions to both hazard model outputs and exposure datasets to i) ensure that input files are projected onto the same WGS84 UTM zone as the target volcano, ii) depending on geographic extent of the input file, either crop it to the extent of the study area or pad it with noData value, and iii) resample the input file to a specified spatial resolution. This preprocessing step produces a set of files with consistent geographic references (i.e. coordinate system, extent and pixel

resolution) and equal numbers of pixels in x and y directions. This step is critical to ensure that the spatial index of pixels is consistent amongst all files, after which exposure is estimated by translating each pixel's spatial index of hazard footprints onto exposure datasets in Numpy. Resampling of the rasters is achieved using a cubic interpolation for continuous hazard data and a nearest neighbor interpolation for discrete exposure data (i.e. cubic and nearest methods of the rasterio.enums.Resampling class, respectively). After resampling, population data are multiplied by the square of the



ratio between original and final resolutions in order to scale population counts to the new pixel surface area. Here, a pixel size of 90 m was adopted to keep computing and storage requirements reasonable while retaining a high enough resolution to allow detailed analysis of exposure. The source code of the GIS framework is available at github.com/vharg/VolcGIS. To support the re-application of our study over space and time, all hazard modelling and exposure assessments were carried out using only open access datasets. Data descriptions and sources are described within each relevant subsection below.

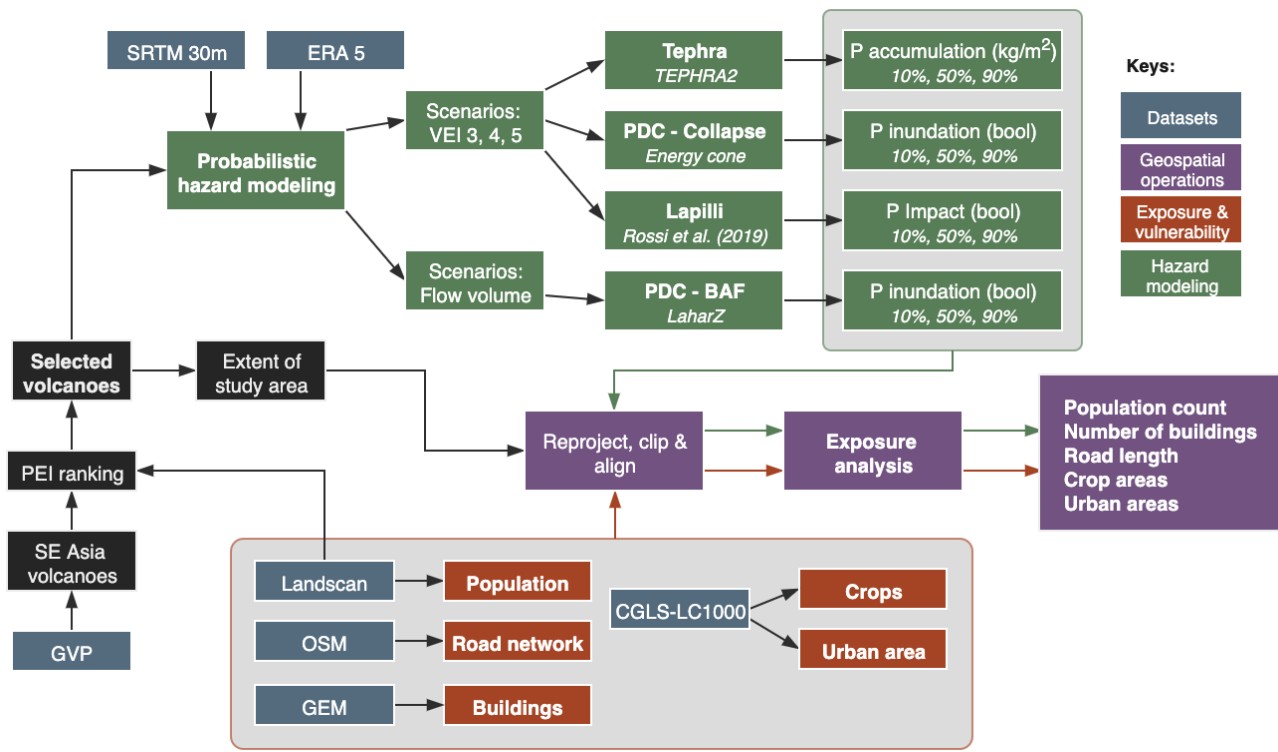

**Figure 2: Schematic outline of the study's methodology for exposure analysis. CGLS-LC: Copernicus Global Land Service-Land Cover; ERA 5: Reanalysis dataset from the European Centre for Medium-Range Weather Forecasts (ECMWF); GEM: Global Earthquake Model; GVP: Global Volcanism Program of the Smithsonian Institute; OSM: OpenStreetMap; PEI: Population Exposure Index; SRTM: Shuttle Radar Topography Mission; bool: Boolean Workflow was made using draw.io.**

## 2.2 Hazard modelling


For each of the 40 volcanoes chosen for hazard modelling, we used openly available hazard models (Appendix A), meteorological data and a DEM to probabilistically simulate potential hazard extent and, where possible, intensity from four explosive volcanic hazards (tephra fall, large clast, dome collapse and column collapse PDC) across three VEI scenarios (VEI 3, 4, and 5). Scenarios were tailored to be generic enough to be applied across all volcanoes whilst

preserving the spatially varying nature of volcanic hazardous phenomena. VEI classes were chosen to span the range of





impacts from explosive volcanic eruptions; VEI 2 eruptions were not simulated because of their limited spatial extent and VEI 6+ eruptions were not simulated as they are of lower probability. However, we do recognise that these scenarios would also be important to consider for comprehensive impact assessments.

Estimating ESPs for regional hazard assessments, especially with such variable eruptive records as those presented by volcanoes in Southeast Asia, is always challenging. With sufficient and consistent knowledge of the eruptive history of selected volcanoes, it could be possible to tailor eruption scenarios to reflect specific types of activity and to use models of increasing complexity (e.g. using 3D numerical tephra dispersal models; Biass et al. 2014). In the face of these data and knowledge gaps, regional hazard assessments targeting volcanoes with differing eruptive histories and record completeness require the development of more generic eruption scenarios that are uniformly assigned to all sources. These eruption scenarios have dominantly been developed around VEI classes (e.g. Jenkins et al. 2012) and, although bypassing the importance of some eruptive processes, they provide key, first-order insights into regional hazard and allow for comparison across multiple scenarios and volcanoes.

The spatial extent (and intensity where possible) of each of our four simulated hazards - tephra fall, large clasts, dome collapse and column collapse PDCs - is quantified using a probabilistic approach designed to account for various epistemic and aleatory sources of uncertainty. The probabilistic approach is implemented either within the model (e.g. column collapse PDC) or by the stochastic sampling of model inputs (e.g. tephra fall). For each hazard and scenario, a generic set of ESPs was developed from global datasets and analogous volcanoes, uniformly applied to all volcanoes used in the study and modelled with a dedicated method (Appendix A). Here, we favour empirical and analytical models over more complex numerical models for two main reasons. Firstly, their relatively lower computing costs allows running probabilistic hazard modelling for all the scenarios and volcanoes and, secondly, they typically require fewer and more generic ESPs. While these models are not necessarily the most physically accurate representation of eruptive processes, they have been shown in numerous circumstances to be acceptable for determining hazard extent and probability (e.g. Tephra2: Bonadonna et al., 2005; Large Clasts: Rossi et al. 2019; PDCs: Ogburn and Calder 2017, Tierz et al. 2016) and were suitable for creating probabilistic hazard inputs for our framework. The next sections describe in more detail the development and the modelling process of eruption scenarios for each hazard, with input parameters and rationalised data sources provided in Appendix A and all hazard outputs in Supplementary Material 1.

### 2.2.1 Tephra fall

Tephra fall is one of the most widespread and frequently occurring volcanic hazards, and can cause damage, disruption or other impacts to buildings, crops, and infrastructure (Jenkins et al., 2015). Here, we simulated the spatial distribution of tephra fall using Tephra2 (Bonadonna et al., 2005), which solves the advection-diffusion-sedimentation equation using a

semi-analytical approach. For each volcano, an eruption scenario was compiled for each of VEI 3, 4 and 5. For each eruption scenario, Tephra2 was run for each of 2,880 synoptic hourly wind profiles (across 10 years) for the wind record closest to the volcano, using a single value of critical ESPs. The variability in the results for each VEI and each volcano is mostly due to the variability in specified wind profiles. The ESPs for each VEI are fixed to ensure convergence in probability and comparable results across volcanoes.

For each volcano and scenario (i.e. VEI), the 2,880 simulations were post-processed to quantify the spatial distribution of probabilities for exceeding a given accumulation of tephra. We chose tephra accumulations that reflect key impacts for our different categories of exposure and follow those defined by Jenkins et al. (2015). A threshold of 1 kg/m$^2$ (approximately equivalent to 1 mm thickness) was used to quantify exposure to people and roads (signifying potential health hazards and disruption to roads). Also, we considered a threshold of 5 kg/m$^2$ (~5 mm) to capture disruption or productivity loss for crops and clean-up and infrastructure disruption in urban areas. Building exposure was quantified using a 100 kg/m$^2$ (~100 mm) threshold, which is often considered as the hazard intensity marking the onset of damage to weak buildings (Jenkins et al., 2014).

Outputs use all 2,880 simulations from the full ten year record to identify the 10%, 50% and 90% probability contours at each of the loading thresholds and VEI scenarios above (27 contours per volcano). Monthly subsets were also extracted to illustrate the variability of hazard as a function of seasonality (an additional 324 contours per volcano). In total, 345,600 individual tephra simulations were processed to produce 4,680 probabilistic outputs (i.e. 360 with aggregated wind conditions and 4,320 for monthly subsets),, with each probabilistic output containing the three probability contours.

### 2.2.2 Large clast

The kinetic energies of lapilli, or large clasts (i.e. particles with diameters of 2-64 mm), produce a dynamic hazard that can cause skull fracture and roof penetration (e.g. Etna 2013, Kelud 2014, Ontake 2014; Williams et al., 2019). As their behaviour is partway between wind-advected particles and ballistics, and because they can be released from the plume margin, large clasts cannot be accurately modelled by models primarily designed for either ballistic trajectory particles or ash dispersal and sedimentation. Here, we used the model of Rossi et al. (2019) that accounts for limited gravitational spreading of the umbrella cloud and the influence of three-dimensional atmospheric conditions on the particles. This model was successfully validated and applied by Osman et al. (2019) to model the extent of coarse lapilli from the 23 November 2013 eruption of Etna.



Here, we considered the threat to human activity in the vicinity of the vent (e.g. hiking activity at the summit). A threshold of kinetic energies ≥30 J at impact was chosen as it represents a central estimate of the impact energy required to cause

skull fracture (Yogandan et al., 1995). This corresponds to a range of clast sizes, depending on density, from ≥3 cm (lithic clasts of 2.5 g/cm$^3$ density) to ≥5.6 cm diameter (pumice clasts of 0.63 g/cm$^3$ density). Thus, we only considered exposure within the extent of the 3 cm lithic isopleth, which is the same extent as a 5.6 cm pumice clast isopleth. The same probabilistic approach was applied for large clast as for tephra fall (i.e. 2,880 wind profiles per volcano and fixed plume heights for each of VEI 3, 4 and 5) to quantify the spatial distribution of impact probabilities by a large clast with a kinetic

energy exceeding 30 J. For each VEI, we extracted isopleth extents associated with 10%, 50% and 90% probabilities (9 outputs in total, per volcano). In total, 345,600 individual simulations were processed to produce 120 probabilistic outputs, with each containing the three probability contours.

### 2.2.3 Dome collapse PDC

PDCs cause more fatal events and fatalities than any other volcanic hazard (Brown et al., 2017). A common mechanism of

PDC generation is the gravitational collapse of a lava dome (Cole et al., 2015). These PDCs are typically valley-confined, but the possible detachment of the dilute component can overspill and inundate populated areas (Lerner et al., under review). We simulated the likely flowpaths of dome collapse PDCs using a recalibrated version of the LAHARZ model (Iverson et al., 1998; Shilling, 1998), with empirical coefficients updated by Widiwijayanti et al. (2009) based on runout and area for dome collapse PDCs at Soufrière Hills, Merapi, Colima and Unzen volcanoes (see Appendix A for more

detail). Since flow volumes are not correlated to VEI, scenarios for our simulations were taken as the volumes corresponding to the 50th (4.5x10$^5$ m$^3$) and 90th percentiles (9.8x10$^6$ m$^3$) obtained from the global dataset, FlowDat (Ogburn, 2012). We did not include the 10th percentile volume (1.1x10$^5$ m$^3$) as it usually results in flows restricted to the summit area. Since flow models generally do not capture PDC overspills or inundation area (as opposed to deposit area) accurately, we applied two buffers around each simulated volume: 300 m and 990 m (1 km rounded to the nearest DEM cell). Buffer

distances were chosen based on extents observed in previous unconfined PDCs (e.g., Merapi 2010, Fuego 2018; Lerner et al., under review). For each simulated volume, we output the 10%, 50% and 90% probabilities for each of the buffer extents. In total, 5,760 individual simulations were processed to produce 160 probabilistic outputs, with each containing the three probability contours.

Although the PDC scenarios and their ESPs were deterministically chosen, we developed a stochastic approach to estimate

the directionality of PDCs from dome collapse. Lava domes often exhibit preferential growth and collapse directions that consequently influence the direction of associated PDCs (Zorn et al., 2019). As factors controlling growth directionality are still debated (e.g. slope and morphology of the summit region; Voight., 2000, Walter et al., 2013),


we developed a new method to automatically identify the travel direction probability for each direction around the
crater based on the summit topography. Although this method is inherently linked to the accuracy of the DEM, it

nonetheless provides a simple, consistent and replicable way to rapidly identify potential flow directions. The
method considers all azimuths - here binned by 10° intervals - around a user-selected release point, and
cumulatively assesses the morphological properties of the crater along a radial distance to express a relative
probability (more details on the method are provided in Appendix B; Tennant et al., in prep). For each volcano the crater
radius was measured using Google Earth and used as the starting point for the flows.  Figure 3 compares the direction

estimated using our method with the reported directions of dome collapse PDCs from four case-study volcanoes and
demonstrates how it successfully captures the dominant flow directions.

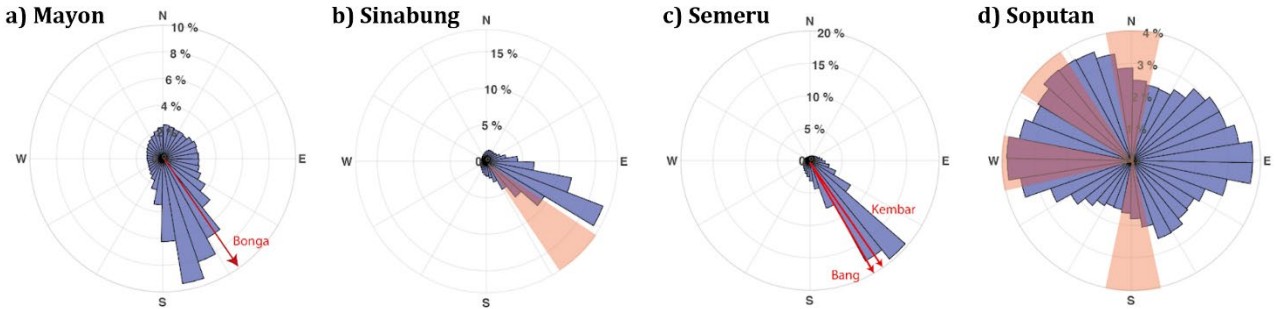

**Figure 3: Probabilistic forecasting of dome collapse PDC travel directions calculated using the SRTM DEM (details in Appendices A and B). Forecasts shown in blue, with actual dome collapse PDC travel directions shown in red for a) Mayon: one**
**dome collapse PDC on 11 June 2001 (GVP, 2001), b)  Sinabung: more than 100 dome collapse PDCs between 30 December 2013 and 4 January 2014 (GVP, 2014), c)  Semeru: several dome collapse PDCs between 30 November and 30 December 2002 (Thouret et al., 2007; Solikhin et al., 2011), and d)  Soputan: several dome collapse PDCs on 1 August 2007 and 25 October 2007 (Kushendratno et al., 2012).  When a single channel was reported this is shown with an arrow accompanied by the channel name, while when only a general direction was reported this is shown as a wedge.**

**2.2.4 Column collapse PDC**

Column collapse events typically produce highly mobile, radially-distributed PDCs (Cole et al., 2015). These are
particularly dangerous, since they are not confined to topographic lows in the same way as other PDCs (e.g. those from
dome collapse). Here, we modelled the PDC inundation using the probabilistic energy cone approach ECMapProb of
Aravena et al. (2020). Following the original approach of Malin and Sheridan (1982), ECMapProb simulates PDC runouts

by projecting a cone with a given height-over-length ratio (H/L) originating from a collapse height onto the topography,
stochastically exploring the uncertainty on collapse height, H/L ratio and vent location. This probabilistic approach allows
a PDC's potential to overcome topographic barriers to be estimated. In doing so, ECMapProb is also able to redistribute



the residual energy after the cone's initial intersection with topography to account for the frequent channelisation of PDCs (Aravena et al. 2020).

Scenarios for column collapse PDCs were defined based on the plume height identified for each VEI, with a collapse height estimated to be ~10% of the plume height (after Wilson et al., 1978). For each VEI scenario, spatial extents of the 10%, 50% and 90% probability of inundation were produced (9 outputs in total, per volcano). More detail on the inputs used are provided in Appendix A. In total, 120 simulations were performed, with each producing a probabilistic output that contained the three probability contours.

## 2.3 Incorporating eruption frequency


The hazard modelling described thus far provides conditional outputs, i.e. they provide the spatial area affected by a given hazard assuming that an eruption of a given VEI has occurred. This is valuable information for crisis planning in the event of unrest; however, comparing across volcanoes at the regional scale requires estimating exposure as a function of the eruption frequency, or probability of occurrence. We achieved this by following the methodology of Hayes et al. (in prep),

which uses a Bayesian update and model combination framework to estimate the annual probability of each VEI and the uncertainty around that value (annual probabilities at the 10%, 50% and 90% probability are provided for each volcano in Supplementary Material SM2). Analogue annual eruption probabilities were first calculated using two volcano analogue classification systems (Whelley et al. 2015; Jenkins et al. 2018). These are then updated using the volcano-specific eruption record sourced from GVP (version 4.8.5, downloaded 20 January 2020). This produces two separate frequency-magnitude

probability distributions, one for each analogue system. These two probability distributions are then combined using a model stacking to produce a single frequency-magnitude probability distribution, with uncertainty. The 50% annual probabilities were used to weight the conditional exposure calculated for each VEI scenario, i.e. each exposure value was multiplied by the annual probability of an eruption of that VEI at that volcano occurring, with the sum across the them providing the averaged annual exposure across all of our eruption simulations and scenarios. This allowed us to better

assess the exposure over given timescales, for example multiplying this weighted exposure by 100 gives the averaged exposure over a 100-year timeframe.

## 2.4 Exposure assessment

Exposure estimates were obtained by overlapping the extent of hazard footprints with exposure datasets within our GIS. We considered five distinct categories of exposure:



1.   Population: The exposure of populations was estimated using Oak Ridge National Laboratory *Landscan* data for 2018. *Landscan* is a proxy for the ambient (i.e. 24h-average) population density at a resolution of ~1 km (Rose et al., 2019).

2.   Number of buildings: The location and number of buildings was modelled using the Global Earthquake Model (GEM) building exposure data described by Silva et al. (2020). Disaggregation of data from the regency level into

built up areas at a 36 by 36 m resolution was achieved using the Pesaresi et al. (2015) Global Human Settlement Layer (GHSL). We considered only residential buildings.

3.   Road lengths and hierarchies: To calculate the length of roads affected by each of our hazards, we used OpenStreet Map (OSM) data (downloaded from Geofrabrik.de on 26 November 2020), which provides the location of roads and their classification, e.g. motorway, primary, residential. We consolidated the 16 OSM road classifications into

four distinct hierarchies: motorway (hierarchy 4), arterial (hierarchy 3), collector (hierarchy 2) and local (hierarchy 1), on the basis that road hierarchy is an indicator of the scale of disruption experienced by the road network from hazardous impacts (Hayes et al., submitted).

4.   Area of crop land: Land cover is used as a proxy for estimating exposure of crops to volcanic hazards. Here we use the Copernicus Global Land Cover v3 at a 100 m resolution (CGLS-LC100: Buchhorn et al., 2020) for 2019

and extract the *cultivated* and *managed vegetation* classes from the discrete classification dataset.

5.   Urban area: As for crops, with the *urban/built-up* class extracted.

All exposure data were interpolated from their original resolution to the 90 x 90 m grid used within our GIS framework, as described in Section 2.1.

## 3 Results

The multi-hazard and multi-exposure analysis presented here required nearly 700,000 individual simulations and produced 26,640 probabilistic outputs (15,240 hazard and 11,400 exposure estimates) that can be useful at the individual volcano scale (e.g. maps of probabilistic dome collapse PDC inundation or the number of buildings exposed to a VEI 4 tephra fall $\geq 1$ kg/m$^2$, at the 10%, 50% and 90% probabilities). We provide all of our hazard and exposure results in the Supplementary Material (SM1, SM2 and SM3). Hazard outputs are provided per volcano and include processed wind direction and speed

information and hazard model outputs. Exposure analysis results are provided as an excel file: these serve as the raw data for all figures and tables in this study. More information on data format for the wind, hazard and exposure data are provided in SM1, SM2 and SM3. These supplemental files include all our data output files, available in user-friendly formats (tif, xlsx).



### 3.1 Case study examples

A total of 498 probabilistic hazard outputs were produced for each volcano (SM1), giving 19,920 in total. Figure 4 highlights three case-study volcanoes, with the reason for choosing each described in the below. We use these as examples of our model outputs and calculated exposure, and the associated hazard and exposure insights that can be derived from our results.

Merapi volcano in central Java, Indonesia, is one of the most active and hazardous volcanoes in the world, with more than

20 million people living within 100 km (Table 2) and more than 20,000 within 10 km (SM3). Our modelling finds that large clasts and dome collapse PDCs are primarily near-vent hazards, with a maximum radial extent of around 7 km to the west for large clasts, and 10 km to the southeast through northwest for dome collapse PDC (Figure 4a). These distances and directionality fit well with deposits produced during the last c.100 years (Voight et al., 2000; Charbonnier and Gertisser, 2008; Jenkins et al., 2016). Results suggest that large clasts and dome collapse PDCs do not affect heavily populated areas,

although transient hiking populations at or near the summit and more heavily populated areas to the northwest (a low probability impact area) are exposed (Figure 4b). Comparison of our model outputs (simulated volume of 9.8 x $10^6$ m$^3$ and buffer extent of 990 m), with mapped 2006 dome collapse PDCs (<2.6 x $10^6$ m$^3$ to the southwest (Ratdomopurbo et al., 2013) and 6 x $10^6$ m$^3$ to the south (Charbonnier and Gertisser, 2008)) show reasonable similarity in runout extent, highlighting the south and southwest as particularly high hazard areas. The comparison also shows that a 30 m-resolution

DEM fails to capture the strong topographic controls evident in mapped PDCs. Note that PDCs during the 2010 eruption extended beyond our simulated PDC footprints to the south by ~5 to 7 km because they were generated by dome explosion and partial column collapse, both of which promote greater runout distances (Jenkins et al. 2013).

Taal volcano, ~60 km to the south of Metro Manila in the Philippines, is a caldera-forming volcano with a history of explosive volcanism (Delos Reyes et al., 2018). More than 25 million people live within 100 km (Table 2) and nearly

60,000 within 10 km (SM3). The strong topographic control of the caldera walls in limiting column collapse PDC runout and exposure at Taal is evident in Figure 4c. Within the caldera scarp, roads are relatively sparse, except for the town of Taal in the southwest where gentler relief results in higher road and population density (Figure 4c,d) and subsequently an increased exposure to topographically controlled hazards such as PDCs. For a VEI 4 scenario, 653 km of predominantly lower hierarchy 1 and 2 roads are exposed to column collapse PDCs at the 10% probability contour, but only 98 km at the

50% probability contour, as PDC runout remains mostly confined to the lake and island. Figure 4d shows the influence of seasonality on the tephra fall impact area, discussed in more detail in Section 4.1. Regardless of season, our modelling shows that ~50 to 60 km of the EH2 highway to the east of the volcano, which links the cities of Batangas and Manila, is likely to be impacted by a VEI 4 eruption from Taal.
Figure 4: Results of hazard modelling for a VEI 4 scenario for a) Large clasts and dome collapse PDCs (9.8 x 10⁶ m³, 990 m buffer) at Merapi, with 2006 dome collapse PDC (Charbonnier et al., 2008) and 2010 large clast extents (Iguchi et al., 2011) shown by the black hashed areas and red dashed radii, respectively; b) Population exposure data at Merapi (LandScan 2018) combined with 50% large clast hazard footprint and the outer limit of dome collapse PDC from (a); c) Column collapse PDC at Taal overlaying the roads, categorised by hierarchy; d) Tephra fall extent at the 50% probability for exceeding 1 kg/m² for January, July and using whole year conditions, overlaying roads, categorised by hierarchy (hierarchy legend as for 4c); e) Crops and urban area exposure at Gede-Pangrango combined with tephra fall probability isopachs for an accumulation of 5 kg/m² for whole year wind conditions; f) Number of buildings at Gede-Pangrango combined with tephra fall accumulation isopachs (50% probability) for whole year wind conditions. Satellite basemap from © Google Maps.


Gede-Pangrango is an active, but recently quiet, volcano with a poorly known eruptive history that lies ~60 km to the south
of Jakarta in western Java (Tennant et al., 2021). This proximity to Jakarta leads to Gede-Pangrango having the greatest
number of people living within 100 km (more than 41 million) of any volcano in our study (Table 2) or the world (Small
and Naumann, 2001). Closer to the volcano, numbers are more modest, with ~15,000 within 10 km (SM3). Figures 4e and
4f show two different ways of plotting our tephra fall results: i) as the probability of exceeding a certain tephra load ($\geq 5$
$kg/m^2$ in Figure 4e), and ii) as the tephra load expected at a given probability (50% in Figure 4f). Both approaches show
that tephra falls are most likely to be dispersed towards the west, affecting only the southernmost parts of Jakarta with
relatively low loads (1 $kg/m^2$). Given a VEI 4 scenario, the city of Sukabumi to the south-southwest and communities to
the west of Gede, along the highway leading into Bogor and Jakarta, are threatened by potentially damaging tephra fall
loads ($\geq 100$ $kg/m^2$: Figure 4f); very atypical wind conditions blowing from the south are needed to result in such loads
across the densely built areas of Jakarta. Considering the low probability scenario (10%) from a VEI 4 eruption, most of
the crops exposed to $\geq 5$ $kg/m^2$ tephra fall accumulation are located to the east of Gede-Pangrango while urban areas are to
the northwest, specifically Bogor (Figure 4e). For the high probability scenario (90%), exposed crops and urban areas are
concentrated within ~20 km to the west of Gede.

### 3.2 Exposure assessment

Each probabilistic hazard output was combined with each of the five exposure datasets to produce 95 exposure estimates
per volcano (3,800 in total: SM3). For most hazards, the exposure increases significantly with increasing VEIs, reflecting
the increased distance reached with greater eruption intensity and/or magnitude (Figures 6, 7, 8). Column collapse PDC
marks the exception, with a VEI 4 or 5 eruption not marking a significant increase in exposure compared to a VEI 3 eruption
(Figure 9). In general, the hazards resulting in the highest values of exposure are, in decreasing order: tephra fallout, PDC
from column collapse, large clasts and PDC from dome collapse. Tephra fall yields a higher population exposure compared
to column collapse PDCs up to accumulations of ~5 $kg/m^2$ for all VEIs. Above a tephra accumulation of 5 $kg/m^2$, column
collapse PDCs result in higher population exposure for a VEI of 3.

### 3.2.1 Population exposure

For all regions and all hazards, the distribution of population exposure is often asymmetrical (positively skewed), with a
long tail suggesting that a smaller number of volcanoes provide the very large exposure values (Figure 5). For tephra fall,
populations in Java are by far the most exposed to our study volcanoes (n=13) of any region (Figures 5 and 6). As the
dominantly east-west wind directions across Java coincide with the island's orientation, tephra is mostly deposited on land.
For an eruption of VEI 5, 12 of the 13 volcanoes in Java result in >10 million people exposed to tephra falls $\geq 1$ $kg/m^2$ from





the volcano; for a VEI 4 scenario, with the exception of Krakatau (~11,000 people), between 3.4 million (Raung) and 9.6 million (Cereme) are exposed to the same tephra fall threshold. Within Java, Krakatau volcano always shows a lower

exposure relative to other Javanese volcanoes, whilst Cereme, due to its upwind location to Jakarta, is consistently amongst the volcanoes resulting in higher exposure to tephra fall. Sulawesi is the region with the second highest median exposure to tephra fall from eruptions with VEI 3, but larger eruptions of VEI≥4 see the Philippines ranked second (Figure 5).

Exposure to large clasts is 3 to 4 orders of magnitude smaller than for tephra fall. Populations in Java and the Philippines have the greatest median exposure to eruptions of VEI 3 and 5 whereas populations in the Halmahera/Banda Sea region

have the greatest median exposure to VEI 4 eruptions (Figure 7). This indicates that our analysis accurately captures the distribution of population in the region, with less people on the flanks of the volcanoes and most settlements being 5-10 km away, often on the shores of volcanic islands. For column collapse PDCs, with a maximum runout distance partway between the maximum extents of large clasts and tephra fall, populations in Java again have the greatest median exposure (Figure 9). For dome collapse PDCs, which typically have a more directed and relatively short maximum extent compared

to the other simulated hazards, median exposure numbers are relatively small but highlight volcanoes in Sumatra (n=2) as those with greatest median exposure and Sulawesi (n=5) as those with the largest exposure values (Figure 8). Lokon-Empung volcano in Sulawesi is driving the larger values in the region (>7,000 people exposed) with the most likely flow direction being to the southeast, affecting communities along the Tomohon-Manado main road, ~5 km away. In Java, Guntur volcano provides the largest outlier exposure value for dome collapse PDCs, with more than 11,000 people exposed

in communities ~7 km southeast from the volcano, on the outskirts of Garut.




**Figure 5: Distribution of the population exposure for each volcano colour-coded by region. The horizontal bars and the coloured circles show the 95% confidence interval and the median, respectively, whereas the small dark dots show the underlying data. Each column is a different eruption scenario (i.e. flow volume for dome collapse PDC, VEI otherwise). The hazard used here considers a conditional probability of occurrence of 50%. The number of volcanoes in each region are as follows: Halmahera/Banda Sea (4), Java (13), Lesser Sunda (10), Sulawesi (5), Sumatra (2), Philippines (6).**

### 3.2.2 Building exposure

For VEI 3, Sulawesi and Sumatra have the largest median number of buildings exposed to tephra accumulations ≥100 kg/m² and Java has the smallest. For VEI≥4, on average, Java becomes the most exposed region with Merapi (VEI 4) and Cereme (VEI 5) producing the largest numbers (Figure 6). Sulawesi and the Philippines are the second two most exposed regions, on average. For large clasts, the regions that have, on average, the most buildings exposed to eruptions of VEI≤4





are Halmahera/Banda Sea, Sulawesi and Sumatra. For VEI≤4, our Javanese volcanoes have virtually no exposed buildings to large clasts, but the region climbs to first place for VEI 5. For column collapse PDCs, the regions with the greatest

median exposure are, in decreasing order: Java, Sulawesi, Sumatra and the Philippines across all VEI classes. For dome collapse PDCs, Sulawesi and Sumatra have the highest median exposure, followed by Halmahera and Java (Figure 8).

### 3.2.3 Road network

Due to the proximity to large and complex urban centres (e.g. Jakarta, Yogyakarta), on average Java has by far the greatest road network exposed to tephra accumulations of ≥1 kg/m$^2$ over all VEIs (Figure 6). For VEI≥4, the region with the second

greatest median exposure is the Philippines, with the notable case of Taal volcano that can affect metropolitan Manila. For VEI 3, only Sumatra and Lesser Sunda have some sections of road (i.e. <20 km) exposed to large clasts. For VEI 4 and 5, the regions with the greatest median exposure are Sumatra and Java, respectively. The pattern of exposure of the road network to column collapse PDC inundation is generally the same as for tephra fall, the only exception being significantly lower median exposure in the Philippines. Due to its location within a caldera lake, Taal volcano requires large eruptions

to affect the road network. Interestingly, the case study of Mayon volcano illustrates the variability of exposure with VEI between tephra fall and PDC. For tephra fall, the main wind direction is westwards, and the urban centre of Legazpi, located ~15 km south-southwest of the vent, becomes increasingly affected by larger eruptions that develop significant crosswind and downwind sedimentation patterns (Figure 6). Conversely, column collapse PDCs are less directional, and the exposed road network varies little across VEIs (Figure 9). Finally, only a limited length of roads (i.e. maximum of 50 km) is typically

exposed to inundation from dome collapse PDCs. For the largest volume and buffer, Guntur and Merapi are the two volcanoes producing the largest road exposure values (Figure 8).

### 3.2.4 Crop area

Regions displaying the largest median exposure of crop land to all hazards are Sumatra and Java. For large clasts and dome collapse PDCs, Sumatra displays the largest median crop exposure across all eruption scenarios (Figure 7 and 9). For large

eruption scenarios (i.e. VEI≥4 for tephra and VEI 5 for column collapse PDC), volcanoes in Java have the largest exposure, on average. The median exposure of crops to tephra accumulations ≥5 kg/m$^2$ in Java varies by two orders of magnitude between VEI 3 (~30 km$^2$) and VEI 5 (~1,700 km$^2$).

### 3.2.5 Urban area

Java and Sulawesi show the highest median exposure to both tephra accumulation ≥1 kg/m$^2$ (Figure 6) and column collapse

PDCs (Figure 9). The third most exposed region, on average, across our volcanoes, is the Philippines for tephra and Sumatra


for column collapse PDCs. Considering a VEI 4 eruption, <1 km² of urban area is exposed, on average, to large clasts in these regions. This increases to 6, 11 and 25 km² for Sumatra, Sulawesi and Java, respectively, for a VEI 5 scenario (Figure 7). The median exposure of urban areas to dome collapse PDC is <2 km² for all regions and scenarios, and is virtually null for Lesser Sundra (Figure 8).

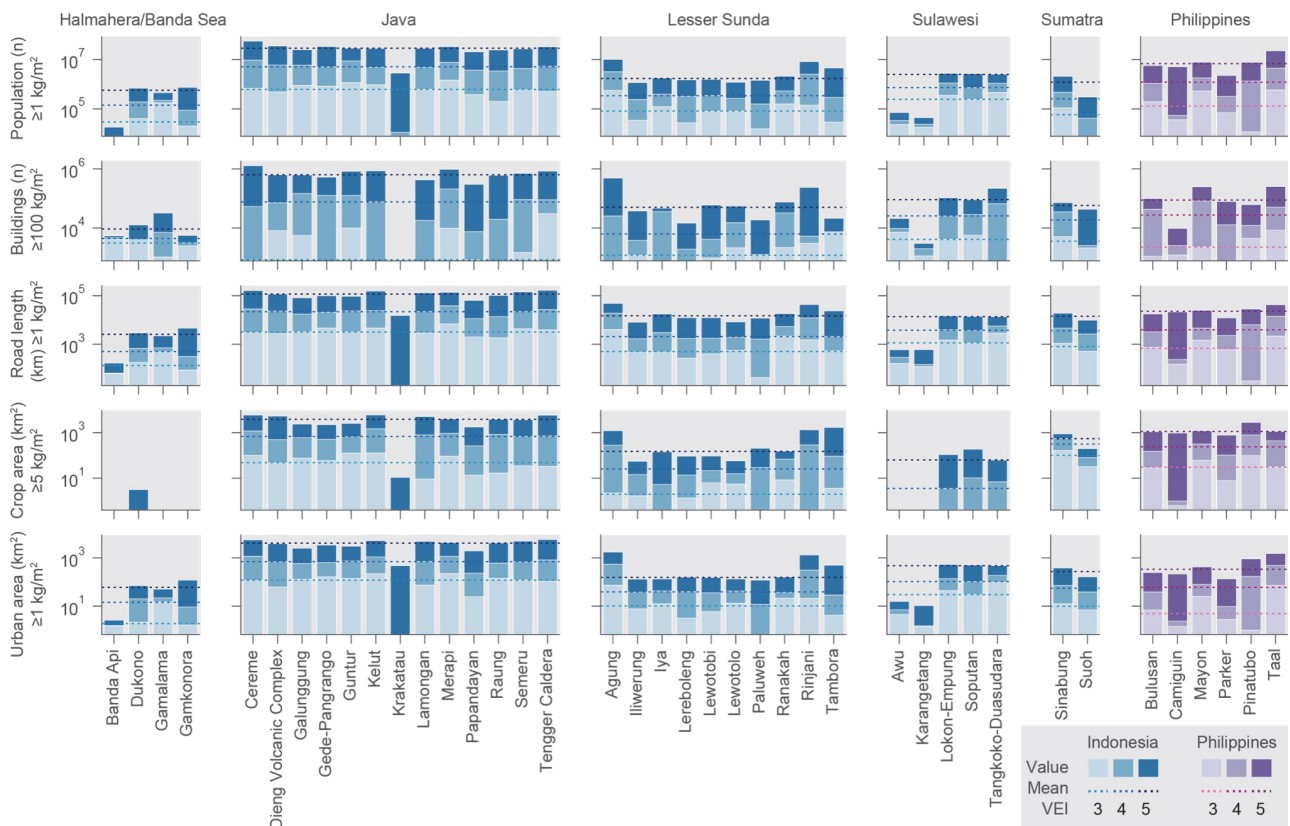

**Figure 6: Exposure to tephra fall accumulation summarised per region and exposure type for a conditional probability of occurrence of 50%. Bars illustrate the variability of exposure with VEI and dotted lines the median for the region. Note that specific thresholds of tephra loads (as defined in section 2.2.1) are used for various exposure types.**





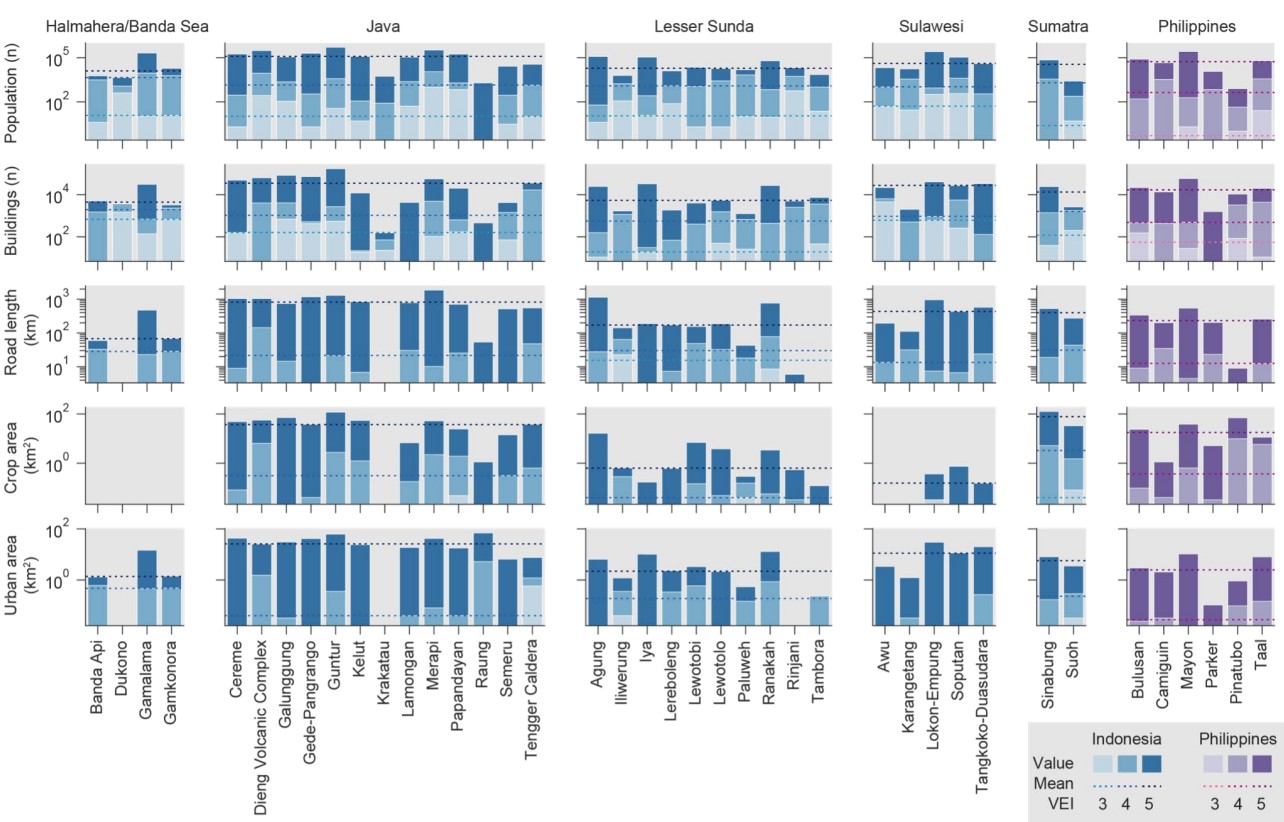

**Figure 7: Exposure to the large clasts hazard (i.e. hazard caused by a kinetic impact ≥30 J) summarised per region and exposure type for a conditional probability of occurrence of 50%. Bars illustrate the variability of exposure with VEI and dotted lines the median for the region.**


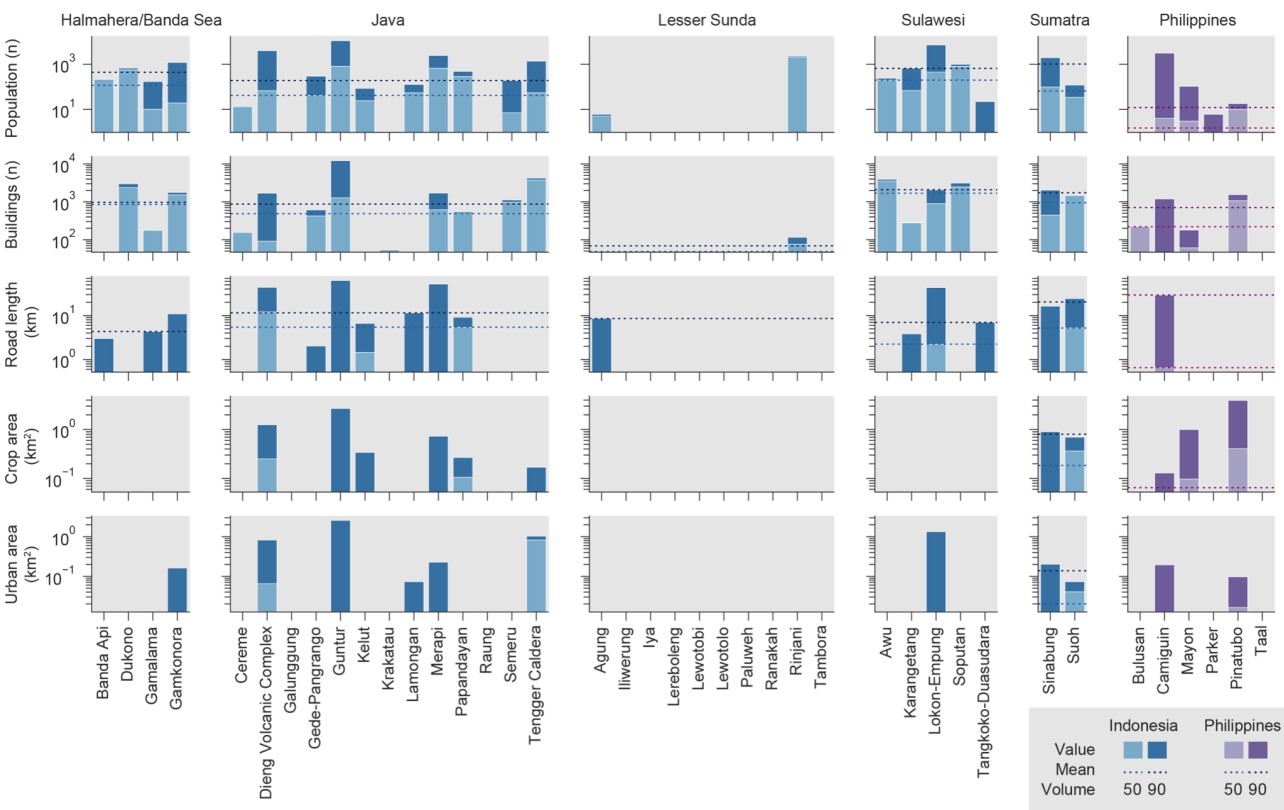

**Figure 8: Exposure to inundation from dome collapse PDCs summarised per region and exposure type. The hazard is extracted for a conditional probability of occurrence of 50% and considers a 990 m buffer around the flow footprint. Bars illustrate the variability of exposure with the initial flow volume and dotted lines the median for the region.**



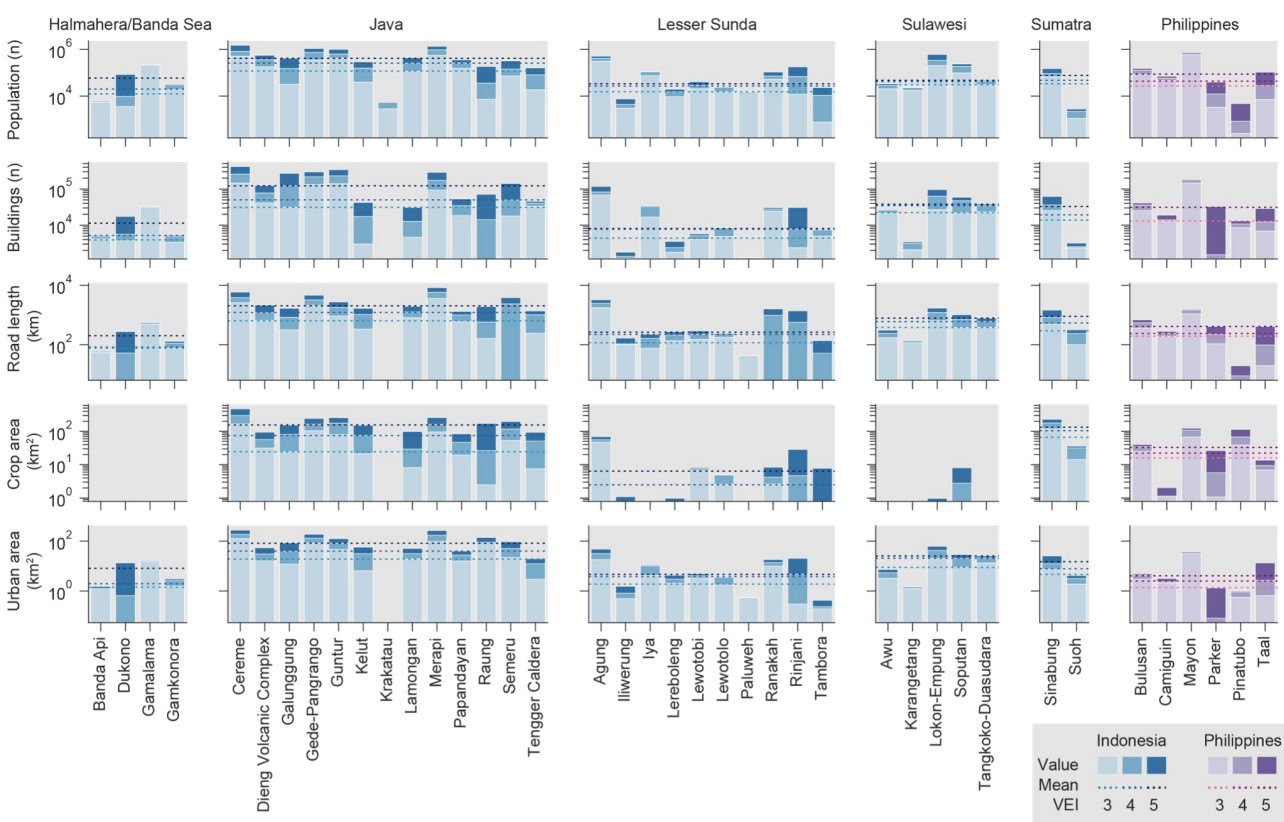

**Figure 9: Exposure to inundation from column collapse PDCs summarised per region and exposure type for a conditional probability of occurrence of 50%. Bars illustrate the variability of exposure with VEI and dotted lines the median for the region.**

## 3.2 Hazard seasonality

Tephra hazard, and related exposure, is strongly controlled by wind conditions at the time of the eruption, which vary across the region as a function of the season. Figure 10 shows the discrepancy between values of population exposure presented above, which aggregate probabilities of tephra fallout over all months, and those calculated using wind conditions from each month separately. We acknowledge two limitations to quantifying our exposure estimates as a function of season.

Firstly, the potential influence of increased rain on hazard modelling (e.g. aggregation increasing proximal sedimentation; Brown et al., 2012) and post-deposition hazard intensity estimates (e.g. increased load due to water-saturated deposits: Williams et al. 2021) is ignored here. Secondly, the population count provided by Landscan is an ambient averaged population, which does not capture any demographical seasonal dynamics (e.g. seasonal workers, tourism etc.).

Three dominant climatic regions exist across our study area (Aldrian and Susanto, 2003): i) Sumatra, Java, Lesser Sunda; ii) Sulawesi, Halmahera/Banda Sea; and iii) the Philippines. Population exposure values for tephra fall from our study




volcanoes in Java generally increase during the peak rainy season (January, February) and decrease during the peak dry season (July, August, September). For the Philippines, the reverse is true with larger population exposure during the peak dry season (January, February, March). Across all of our study volcanoes, relative changes to population exposure estimates as a result of seasonal variability are typically constrained to within ± 10% of the whole year estimate (Figure 10).

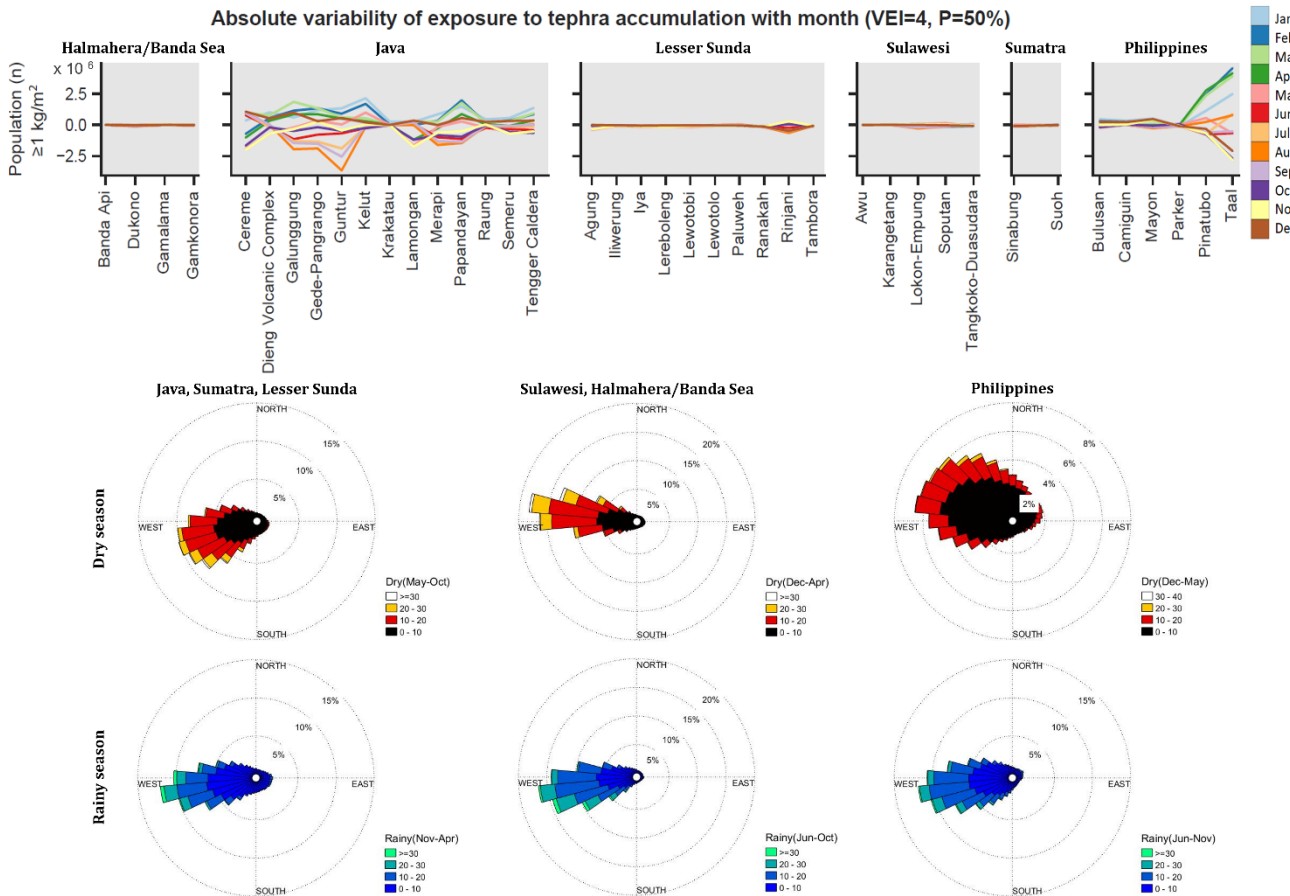

**Figure 10: Seasonality patterns in population exposure (top row) from the 50% probability of tephra fallout ≥ 1 kg/m$^2$ associated with a VEI 4 eruption. Values represent the difference in exposure value between the tephra hazard estimated from all 2,880 hourly synoptic wind profiles across the whole 12 months of a year (normalised to the 0 line) and a subset of this total population of wind profiles extracted per month, where each line represents a different month. A value lower than zero represents a decreased exposure in that month and a value greater than zero the opposite. Averaged wind conditions for the dry and rainy seasons for the three main climatic regions are shown in the next rows for altitudes of 5 to 15 km above sea level.**

Three volcanoes in the region best illustrate changes in population exposure as a function of the month of the eruption. Firstly, an eruption at Krakatau volcano in January leads to a drastic increase in population exposure compared to the rest of the year. Considering a VEI 4 eruption and a 50% probability of occurrence, an eruption in January leaves ~270,000



people exposed to an accumulation $\geq 1$ kg/m$^2$ compared to ~10,000 when all months are aggregated. Wind conditions in January reduce the westward extent of the $\geq 1$ kg/m$^2$ isopach and extend it eastwards, affecting human settlements on the western parts of Java, whilst dispersal for other months generally extends the isopach towards the southwest, resulting in deposition in the Sunda strait. A similar behaviour is observed at Guntur volcano; although the dominant wind direction is towards the southwest, winds during the rainy months (Dec-Apr) also display dispersal towards the north and the east,

which increase the probability of Bandung (9 million people, northwest of Guntur) and Garut (100,000 people, southeast of Guntur) being affected by $\geq 1$ kg/m$^2$ of tephra. Finally, winds at Taal volcano show a strong northward component around the tropopause (~8 to 15 km) during the peak dry season (e.g. January) compared to the rest of the year, when winds at this height mostly blow towards the west. As a result, eruptions during the month of January increase the probability of tephra deposits affecting Metro Manila, as demonstrated by Taal's January 2020 eruption.

**4 Volcano ranking**

The multi-hazard and multi-exposure analysis presented here allows us to rank all 40 volcanoes according to their exposure to volcanic hazards (Figure 11). The ranking is performed separately for each hazard and exposure type and simply reflects the relative rank of the computed exposure in decreasing order. Separate rankings are presented per VEI scenario, providing 55 "short-term" (i.e. conditional to the occurrence of the eruption scenario) estimates and 15 "long-term" scale (i.e.

accounting for the probability of occurrence of the eruption scenario: Section 3.2) insights for each volcano. Aggregated results are shown here for each hazard separately (Figures 12 through 15), with individual volcano results provided in Supplementary Material 3.

The five volcanoes that rank the highest overall (Merapi, Guntur, Dieng, Cereme, Gede-Pangrango: Figure 11) include ranks that range between 1 and 38 (out of 40), showing the wide variability in exposure when multiple hazards, scenarios

and exposure categories are considered. Raung, Suoh and Pinatubo exhibit ranks across the full range; for example, Pinatubo ranks as the volcano with the greatest exposure of crop areas to dome collapse PDCs and large clasts from a VEI 4 scenario *and* as the volcano with the smallest exposure of population to VEI 3 and 4 column collapse PDCs and VEI 5 large clasts, with other permutations falling between rank 4 and 39 (Figure 11a).



**Figure 11: upper)** Heat map to show the number of times (cell colour) that a certain rank (x axis) is assigned for each volcano (y axis) for the 55 ranking permutations across hazard, VEI/volume, and exposure; **lower)** Individual ranks and exposure estimates for our highest overall ranked volcano, Merapi. The y-axis and text numbers shows the rank for each combination of hazard (columns), VEI or volume (rows) and exposure (colour), with the size of the circle reflecting the exposure values normalised to the largest value for that combination from any of our 40 study volcanoes; the black circle represents this largest value. For example, for a VEI 3 tephra fall from Merapi, while buildings rank higher (3) than crop area (6), the exposure value for crop area is closer to the maximum calculated across all of our volcanoes than it is for buildings.


Merapi is the only one of our 40 study volcanoes to remain within the top five ranked volcanoes for population exposure across all hazards, all VEI scenarios and for both conditional probabilities and those incorporating eruption frequency (Figure 11b; Supplementary Material 3). For other exposure categories, Merapi remains within the top six of all volcanoes

for the more distal tephra fall and column collapse PDC hazards. For the more proximal hazards of large clasts and dome collapse PDCs, there is large variation within the lower VEI and volume scenarios while the higher VEI and volume scenarios all give ranks within the top nine. For example, building exposure to VEI 4 large clast and lower volume dome collapse PDC scenarios give ranks of 24 and 23, respectively, while the same exposure for VEI 5 large clast and upper volume dome collapse PDC results in ranks of 1 and 2, respectively (Figure 11b). This supports our earlier finding (Section

3.1) that large clasts and dome collapse PDCs are less likely to affect heavily populated areas unless the eruption is large, although exposure estimates are still higher than for most of our study volcanoes.

Gede-Pangrango, a stratovolcano ~60 km to the south of Jakarta, ranks as having high population exposure when radii are assumed (Small and Naumann, 2001; Table 2). For the more distal hazards of relatively thin ($\geq 1$ or $\geq 5$ kg/m$^2$) tephra fall (Figure 12) and column collapse PDC (Figure 15a), this mostly holds true (ranked within the top 12 for all but building

exposure to tephra falls $\geq 100$ kg/m$^2$ from a VEI 3 scenario, which is rank 31). For the more proximal large clast and dome collapse PDC hazards, Gede-Pangrango for the most part ranks relatively low for all exposure categories (14 to 37), with the exception of the VEI 5 large clast scenario (ranks 5 through 10). Large clasts typically fall within a 10 km radius for the VEI 5 scenario at Gede-Pangrango, meaning that they affect the outskirts of a number of towns, e.g. Cibodas to the northeast, and associated cropland that rises up the valleys between the towns and the volcano (Figure 4e). Ranks are

generally lower for the absolute, rather than conditional, estimates, reflecting a relatively low eruption frequency compared to other case-study volcanoes. These findings highlight that while Gede-Pangrango has previously been considered the volcano with the highest population exposure in the world (Table 2), this is not the case when likely hazard footprints and eruption probabilities are taken into account: while exposure remains high for the more distal hazards, for more proximal hazards, other volcanoes in our study pose a greater threat.

Closed-vent systems (sealed conduit), such as Gede-Pangrango, Guntur and Cereme, are more likely to produce large explosive eruptions (Bebbington, 2014), and these are exactly the volcanoes that we want to highlight with our approach: those that may be currently quiet but that have the potential to cause significant impact when they reawaken. This study provides a preliminary assessment of areas, populations and assets that may be affected in a future eruption from such volcanoes, highlighting hotspots where there could be a relatively large impact. Guntur is one such volcano as it lies ~35

km southeast of the second largest metropolitan area in Indonesia, Bandung and ~10 km northwest of the town of Garut and hosts abundant crop areas on the plains around the volcano. Guntur is a complex of overlapping stratovolcanoes, with the youngest cone having produced frequent explosive eruptions (VEI 2-3) between 1800 and 1847 (n=21), making it one

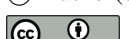


of the most active volcanoes in the study area during this time; however, there have been no eruptions since 1847 (Global

Volcanism Program, 2013) suggesting that the eruptive regime has changed from that of an open-vent to a closed-vent system. Exposure around Guntur is particularly high for modelled dome collapse and column collapse PDCs and for VEI 5 large clast impact, as hazard footprints reach the outskirts of Garut. For tephra fall, Guntur is ranked within the top 10 volcanoes for all exposure categories and VEI scenarios, with the rank typically decreasing with increasing VEI as volcanoes with larger distal downwind populations begin to dominate the rankings.

| | Population (n) exposed to ≥1 kg/m² | | | | Buildings (n) exposed to ≥100 kg/m² | | | | Road length (km) exposed to ≥1 kg/m² | | | | Crop areas (km2) exposed to ≥5 kg/m² | | | | Urban areas (km2) exposed to ≥1 kg/m² | | | |
|---|---|---|---|---|---|---|---|---|---|---|---|---|---|---|---|---|---|---|---|---|
| VEI | 3 | 4 | 5 | A | 3 | 4 | 5 | A | 3 | 4 | 5 | A | 3 | 4 | 5 | A | 3 | 4 | 5 | A |
| Merapi | 1 | 3 | 4 | 1 | 3 | 1 | 2 | 1 | 1 | 1 | 5 | 1 | 6 | 3 | 6 | 2 | 1 | 2 | 6 | 1 |
| Tengger Caldera | 12 | 6 | 5 | 2 | 1 | 6 | 4 | 2 | 6 | 3 | 1 | 2 | 12 | 7 | 3 | 1 | 9 | 4 | 1 | 2 |
| Cereme | 6 | 1 | 1 | 12 | 30 | 11 | 1 | 11 | 9 | 2 | 2 | 12 | 4 | 2 | 2 | 11 | 8 | 1 | 2 | 10 |
| Guntur | 2 | 2 | 6 | 7 | 2 | 3 | 5 | 5 | 8 | 5 | 10 | 9 | 3 | 9 | 10 | 9 | 5 | 5 | 10 | 9 |
| Kelut | 3 | 9 | 8 | 9 | 37 | 8 | 3 | 7 | 2 | 4 | 3 | 6 | 2 | 1 | 1 | 6 | 2 | 3 | 3 | 6 |
| Dieng Volcanic Complex | 13 | 4 | 2 | 6 | 5 | 9 | 7 | 8 | 12 | 6 | 7 | 7 | 10 | 13 | 4 | 7 | 14 | 7 | 8 | 7 |
| Semeru | 8 | 11 | 9 | 3 | 21 | 5 | 6 | 3 | 4 | 7 | 4 | 3 | 11 | 8 | 8 | 4 | 7 | 8 | 4 | 3 |
| Galunggung | 4 | 5 | 10 | 13 | 9 | 2 | 8 | 13 | 7 | 11 | 11 | 14 | 7 | 10 | 11 | 15 | 6 | 11 | 11 | 14 |
| Gede-Pangrango | 5 | 7 | 3 | 8 | 31 | 4 | 10 | 9 | 3 | 9 | 9 | 8 | 9 | 12 | 12 | 10 | 3 | 9 | 9 | 8 |
| Lamongan | 7 | 8 | 7 | 5 | 34 | 21 | 12 | 10 | 10 | 8 | 6 | 5 | 18 | 6 | 5 | 5 | 11 | 6 | 5 | 5 |
| Taal | 9 | 10 | 12 | 10 | 4 | 12 | 14 | 12 | 13 | 12 | 15 | 11 | 14 | 14 | 18 | 12 | 13 | 13 | 14 | 11 |
| Raung | 18 | 13 | 11 | 4 | 39 | 20 | 9 | 4 | 16 | 13 | 8 | 4 | 16 | 4 | 7 | 3 | 4 | 10 | 7 | 4 |
| Agung | 10 | 14 | 14 | 19 | 33 | 18 | 11 | 15 | 5 | 10 | 13 | 16 | 24 | 17 | 16 | 17 | 12 | 12 | 13 | 16 |
| Mayon | 11 | 16 | 16 | 11 | 17 | 7 | 15 | 6 | 19 | 20 | 17 | 10 | 8 | 15 | 17 | 8 | 17 | 20 | 22 | 12 |
| Papandayan | 15 | 12 | 13 | 15 | 29 | 26 | 13 | 21 | 15 | 15 | 12 | 19 | 17 | 18 | 13 | 18 | 18 | 15 | 12 | 19 |
| Rinjani | 22 | 15 | 15 | 14 | 15 | 29 | 16 | 17 | 17 | 14 | 14 | 13 | 30 | 16 | 15 | 14 | 20 | 14 | 15 | 13 |
| Sinabung | 24 | 23 | 27 | 32 | 10 | 15 | 23 | 28 | 21 | 19 | 20 | 30 | 1 | 11 | 21 | 19 | 23 | 21 | 23 | 28 |
| Pinatubo | 37 | 17 | 17 | 23 | 11 | 24 | 24 | 30 | 39 | 16 | 16 | 27 | 5 | 5 | 9 | 16 | 38 | 17 | 16 | 23 |
| Tangkoko-Duasudara | 14 | 19 | 24 | 25 | 36 | 10 | 17 | 20 | 11 | 17 | 27 | 25 | 28 | 30 | 31 | 33.5 | 10 | 16 | 20 | 21 |
| Bulusan | 19 | 20 | 18 | 16 | 26 | 13 | 19 | 18 | 22 | 23 | 22 | 20 | 15 | 19 | 19 | 13 | 26 | 26 | 24 | 22 |
| Lokon-Empung | 16 | 18 | 22 | 18 | 12 | 19 | 18 | 16 | 18 | 21 | 25 | 17 | 33 | 32 | 28 | 24.5 | 15 | 18 | 17 | 17 |
| Soputan | 17 | 21 | 23 | 17 | 8 | 17 | 20 | 14 | 20 | 22 | 26 | 15 | 31 | 29 | 25 | 22 | 16 | 19 | 18 | 15 |
| Ranakah | 21 | 22 | 26 | 26 | 18 | 16 | 22 | 23 | 14 | 18 | 21 | 23 | 19 | 23 | 26 | 24.5 | 19 | 28 | 27 | 30 |
| Tambora | 31 | 28 | 20 | 28 | 7 | 27 | 31 | 32 | 25 | 27 | 18 | 29 | 23 | 21 | 14 | 20 | 30 | 30 | 19 | 26 |
| Iya | 23 | 24 | 28 | 27 | 32 | 14 | 27 | 24 | 28 | 24 | 23 | 24 | 32 | 31 | 27 | 28 | 24 | 22 | 32 | 28 |
| Lewotolo | 25 | 29 | 32 | 30 | 19 | 22 | 26 | 26 | 24 | 28 | 33 | 31 | 22 | 26 | 32 | 31 | 22 | 23 | 31 | 28 |
| Parker | 27 | 27 | 25 | 37 | 40 | 23 | 21 | 38 | 23 | 26 | 30 | 39 | 20 | 20 | 22 | 30 | 32 | 34 | 30 | 39 |
| Lewotobi | 26 | 25 | 29 | 22 | 28 | 31 | 25 | 21 | 30 | 29 | 28 | 21 | 21 | 25 | 29 | 23 | 28 | 29 | 29 | 24 |
| Suoh | 38 | 36 | 37 | 39 | 20 | 35 | 28 | 35 | 26 | 25 | 32 | 36 | 13 | 22 | 24 | 27 | 27 | 25 | 26 | 35.5 |
| Iliwerung | 30 | 30 | 33 | 29 | 23 | 33 | 29 | 27 | 29 | 31 | 34 | 26 | 25 | 27 | 33 | 29 | 25 | 27 | 33 | 25 |
| Camiguin | 29 | 35 | 19 | 24 | 22 | 36 | 36 | 37 | 34 | 37 | 19 | 28 | 27 | 33 | 20 | 21 | 36 | 37 | 25 | 32 |
| Leroboleng | 32 | 26 | 30 | 33 | 35 | 38 | 34 | 39 | 31 | 30 | 29 | 33 | 26 | 28 | 30 | 32 | 31 | 24 | 28 | 33 |
| Gamalama | 20 | 31 | 36 | 21 | 27 | 28 | 30 | 19 | 27 | 33 | 37 | 22 | 35.5 | 36.5 | 36 | 37.5 | 21 | 31 | 37 | 20 |
| Paluweh | 36 | 33 | 31 | 31 | 24 | 39 | 33 | 34 | 38 | 32 | 31 | 32 | 29 | 24 | 23 | 26 | 39 | 33 | 35 | 34 |
| Dukono | 28 | 32 | 35 | 36 | 14 | 32 | 35 | 36 | 32 | 34 | 36 | 38 | 38.5 | 34 | 35 | 37.5 | 33 | 32 | 36 | 38 |
| Awu | 33 | 37 | 38 | 38 | 6 | 25 | 32 | 25 | 33 | 36 | 39 | 37 | 35.5 | 36.5 | 39 | 37.5 | 29 | 36 | 38 | 37 |
| Gamkonora | 34 | 34 | 34 | 34 | 16 | 34 | 37 | 33 | 36 | 35 | 35 | 35 | 38.5 | 39 | 37 | 37.5 | 34 | 35 | 34 | 31 |
| Krakatau | 40 | 39 | 21 | 20 | 38 | 40 | 40 | 40 | 40 | 40 | 24 | 18 | 38.5 | 39 | 34 | 33.5 | 40 | 40 | 21 | 18 |
| Karangetang | 35 | 38 | 39 | 35 | 25 | 37 | 39 | 29 | 35 | 38 | 38 | 34 | 34 | 35 | 38 | 37.5 | 37 | 39 | 39 | 35.5 |
| Banda Api | 39 | 40 | 40 | 40 | 13 | 30 | 38 | 31 | 37 | 39 | 40 | 40 | 38.5 | 39 | 40 | 37.5 | 35 | 38 | 40 | 40 |





**Figure 12: Individual rankings for calculated exposure using the 50% tephra fall hazard for all volcanoes across the five exposure categories. Columns '3', '4', '5' quantify the short-term exposure to the conditional occurrence of VEI 3, 4 and 5 scenarios. Column 'A' quantifies the long-term exposure, using the sum of all scenarios, where each scenario was weighted by its probability of occurrence. For each individual column, all volcanoes are attributed a rank between 1–40, where 1 is considered the highest (i.e. the largest exposure; dark red cells) and 40 the lowest (dark blue cells). Volcanoes are ordered from highest to lowest ranking**

**across all conditional categories (equal weighting assumed). For instance, Raung volcano is the 11th ranked volcano for population exposure to ≥1 kg/m² when considering a VEI 5 eruption (with the highest and lowest being Cereme and Banda Api, respectively), but is amongst the volcanoes with the lowest rank (39) for building exposure to 100 kg/m² when considering a VEI 3 eruption.**

| VEI | Population (n) 3 | 4 | 5 | A | Buildings (n) 3 | 4 | 5 | A | Road length (km) 3 | 4 | 5 | A | Crop areas (km²) 3 | 4 | 5 | A | Urban areas (km²) 3 | 4 | 5 | A |
|---|---|---|---|---|---|---|---|---|---|---|---|---|---|---|---|---|---|---|---|---|
| Guntur | 13 | 8 | 1 | 1 | 6 | 11 | 1 | 4 | 22 | 18 | 2 | 2 | 5 | 5 | 2 | 5 | 22 | 9.5 | 2 | 3 |
| Dieng Volcanic Complex | 7 | 3 | 3 | 5 | 34.5 | 7 | 4 | 5 | 22 | 1 | 5 | 5 | 23 | 2 | 5 | 2 | 22 | 2 | 8 | 7.5 |
| Merapi | 1 | 1 | 2 | 3 | 14 | 4 | 6 | 1 | 22 | 24 | 1 | 1 | 23 | 6 | 7 | 1 | 22 | 19 | 4 | 2 |
| Papandayan | 2 | 17 | 9 | 22 | 10 | 23 | 19 | 15 | 22 | 14 | 12 | 15.5 | 2 | 7 | 13 | 22 | 22 | 20.5 | 12 | 18.5 |
| Galunggung | 9 | 16 | 12 | 11 | 3 | 6 | 2 | 20 | 22 | 21 | 11 | 10.5 | 23 | 34.5 | 3 | 19 | 22 | 23 | 6 | 15.5 |
| Tengger Caldera | 20.5 | 19 | 22 | 2 | 34.5 | 1 | 9 | 12 | 22 | 5 | 14 | 4 | 23 | 11 | 11 | 3 | 1 | 3 | 20 | 7.5 |
| Suoh | 23.5 | 33 | 38 | 36 | 9 | 14 | 33 | 39 | 3 | 6 | 21 | 18 | 1 | 8 | 12 | 32.5 | 3 | 12 | 23 | 34 |
| Lokon-Empung | 6 | 25 | 5 | 9 | 5 | 20 | 8 | 6 | 22 | 27 | 7 | 31.5 | 4 | 24 | 29 | 9 | 22 | 33 | 7 | 6 |
| Sinabung | 37 | 10 | 17 | 26 | 19 | 18 | 16 | 26 | 22 | 19 | 16 | 8.5 | 23 | 4 | 1 | 25 | 22 | 15 | 18 | 24 |
| Ranakah | 22 | 26 | 18 | 17 | 34.5 | 29 | 13 | 21 | 2 | 2 | 10 | 21 | 23 | 19 | 22 | 14 | 22 | 4 | 14 | 18.5 |
| Taal | 15 | 9 | 19 | 12 | 25.5 | 5 | 20 | 11 | 22 | 23 | 22 | 10.5 | 23 | 3 | 17 | 18 | 22 | 16.5 | 19 | 15.5 |
| Gamalama | 19 | 2 | 6 | 6 | 13 | 21 | 12 | 3 | 22 | 17 | 18 | 31.5 | 23 | 34.5 | 35.5 | 6 | 22 | 8 | 13 | 5 |
| Soputan | 5 | 7 | 15 | 8 | 8 | 3 | 14 | 8 | 22 | 30 | 19 | 31.5 | 23 | 34.5 | 25 | 12 | 22 | 33 | 15 | 12.5 |
| Gede-Pangrango | 30 | 29 | 7 | 7 | 7 | 24 | 3 | 7 | 22 | 36.5 | 3 | 7 | 23 | 21.5 | 10 | 8 | 22 | 33 | 5 | 4 |
| Cereme | 30 | 30 | 8 | 14 | 11 | 31 | 7 | 14 | 22 | 26 | 6 | 13.5 | 23 | 18 | 8 | 15 | 22 | 33 | 3 | 12.5 |
| Iliwerung | 8 | 18 | 34 | 34 | 28 | 19 | 36 | 33 | 1 | 3 | 30 | 31.5 | 23 | 13 | 26 | 20 | 2 | 9.5 | 33 | 34 |
| Lamongan | 11 | 15 | 14 | 24 | 34.5 | 39.5 | 28 | 9 | 22 | 11 | 9 | 15.5 | 23 | 14 | 19 | 4 | 22 | 20.5 | 11 | 9 |
| Mayon | 30 | 34 | 4 | 4 | 20 | 26 | 5 | 2 | 22 | 31 | 15 | 3 | 23 | 10 | 9 | 7 | 22 | 33 | 16 | 10.5 |
| Kelut | 23.5 | 36 | 10 | 18 | 23.5 | 37 | 22 | 10 | 22 | 29 | 8 | 6 | 23 | 9 | 6 | 11 | 22 | 33 | 9 | 10.5 |
| Awu | 12 | 23 | 25 | 10 | 1 | 2 | 17 | 27 | 22 | 22 | 25 | 31.5 | 23 | 27.5 | 35.5 | 24 | 22 | 33 | 24 | 24 |
| Gamkonora | 17.5 | 5 | 27 | 23 | 4 | 13 | 32 | 25 | 22 | 12 | 32 | 31.5 | 23 | 34.5 | 38.5 | 31 | 22 | 7 | 30 | 34 |
| Lewotobi | 30 | 22 | 24 | 29 | 27 | 30 | 30 | 23 | 22 | 4 | 29 | 18 | 23 | 15.5 | 18 | 21 | 22 | 6 | 25 | 24 |
| Lewotolo | 30 | 14 | 28 | 30 | 17 | 15 | 25 | 31 | 22 | 9 | 27 | 21 | 23 | 20 | 21 | 26.5 | 22 | 33 | 28 | 34 |
| Paluweh | 20.5 | 4 | 30 | 35 | 21 | 22 | 38 | 29 | 22 | 20 | 35 | 31.5 | 3 | 15.5 | 30 | 34 | 22 | 16.5 | 35 | 34 |
| Agung | 25.5 | 38 | 11 | 20 | 25.5 | 32 | 15 | 18 | 22 | 13 | 4 | 18 | 23 | 34.5 | 15 | 13 | 22 | 33 | 22 | 24 |
| Bulusan | 37 | 35 | 16 | 13 | 12 | 27 | 18 | 13 | 22 | 25 | 20 | 12 | 23 | 17 | 14 | 17 | 22 | 25 | 26 | 24 |
| Semeru | 27 | 31 | 23 | 15 | 16 | 17 | 29 | 16 | 22 | 36.5 | 17 | 8.5 | 23 | 12 | 16 | 10 | 22 | 33 | 21 | 14 |
| Camiguin | 37 | 12 | 20 | 28 | 34.5 | 28 | 21 | 24 | 22 | 7 | 24 | 31.5 | 23 | 21.5 | 23.5 | 28 | 22 | 23 | 29 | 34 |
| Tangkoko-Duasudara | 37 | 28 | 21 | 19 | 34.5 | 33 | 10 | 28 | 22 | 15 | 13 | 31.5 | 23 | 34.5 | 32 | 23 | 22 | 13 | 10 | 18.5 |
| Pinatubo | 33 | 39 | 40 | 31 | 15 | 10 | 23 | 40 | 22 | 36.5 | 36 | 13.5 | 23 | 1 | 4 | 36.5 | 22 | 18 | 34 | 34 |
| Banda Api | 25.5 | 13 | 35 | 27 | 34.5 | 16 | 27 | 32 | 22 | 8 | 33 | 31.5 | 23 | 34.5 | 38.5 | 30 | 22 | 5 | 31 | 24 |
| Karangetang | 14 | 11 | 29 | 25 | 34.5 | 25 | 34 | 19 | 22 | 10 | 31 | 31.5 | 23 | 6 | 34 | 16 | 22 | 23 | 32 | 24 |
| Rinjani | 3 | 6 | 26 | 21 | 34.5 | 12 | 26 | 22 | 22 | 36.5 | 37 | 31.5 | 23 | 24 | 28 | 36.5 | 22 | 33 | 39 | 34 |
| Iya | 17.5 | 32 | 13 | 16 | 23.5 | 36 | 11 | 17 | 22 | 32 | 26 | 31.5 | 23 | 34.5 | 31 | 29 | 22 | 33 | 17 | 18.5 |
| Tambora | 16 | 24 | 33 | 33 | 18 | 9 | 24 | 36 | 22 | 36.5 | 39 | 31.5 | 23 | 34.5 | 33 | 39 | 22 | 14 | 36 | 34 |
| Leroboleng | 10 | 20 | 31 | 39 | 34.5 | 35 | 35 | 34 | 22 | 28 | 28 | 31.5 | 23 | 34.5 | 27 | 32.5 | 22 | 11 | 27 | 34 |
| Dukono | 4 | 21 | 37 | 32 | 2 | 8 | 31 | 37 | 22 | 36.5 | 39 | 31.5 | 23 | 34.5 | 38.5 | 39 | 22 | 33 | 39 | 34 |
| Raung | 37 | 40 | 39 | 37 | 34.5 | 38 | 39 | 35 | 22 | 36.5 | 34 | 21 | 23 | 27.5 | 23.5 | 26.5 | 22 | 1 | 1 | 1 |
| Parker | 37 | 27 | 32 | 40 | 34.5 | 39.5 | 37 | 38 | 22 | 16 | 23 | 31.5 | 23 | 24 | 20 | 35 | 22 | 33 | 37 | 34 |
| Krakatau | 37 | 37 | 36 | 38 | 22 | 34 | 40 | 30 | 22 | 36.5 | 39 | 31.5 | 23 | 34.5 | 38.5 | 39 | 22 | 33 | 39 | 34 |

**Figure 13: Individual rankings for calculated exposure using the 50% large clast hazard. Column names, volcano order and cell colour as for Figure 12.**




| | Population (n) | | Buildings (n) | | Road length (km) | | Crop areas (km²) | | Urban areas (km²) | |
|---|---|---|---|---|---|---|---|---|---|---|
| Flow volume (m³): | 4.5E+05 | 9.8E+06 | 4.5E+05 | 9.8E+06 | 4.5E+05 | 9.8E+06 | 4.5E+05 | 9.8E+06 | 4.5E+05 | 9.8E+06 |
| Dieng Volcanic Complex | 12 | 3 | 19 | 10 | 1 | 3 | 3 | 3 | 2 | 4 |
| Guntur | 3 | 1 | 7 | 1 | 23.5 | 1 | 6.5 | 2 | 22.5 | 1 |
| Lokon-Empung | 6 | 2 | 10 | 6 | 4 | 4 | 6.5 | 12 | 22.5 | 2 |
| Suoh | 16 | 20 | 6 | 12 | 3 | 6 | 2 | 7 | 3 | 10.5 |
| Merapi | 4 | 5 | 11 | 9 | 23.5 | 2 | 10 | 6 | 22.5 | 5 |
| Tengger Caldera | 13.5 | 8 | 1 | 2 | 23.5 | 29 | 25.5 | 10 | 1 | 3 |
| Papandayan | 7 | 13 | 12 | 16 | 2 | 10 | 4 | 9 | 22.5 | 26 |
| Sinabung | 10 | 7 | 13 | 7 | 23.5 | 7 | 25.5 | 5 | 22.5 | 6 |
| Pinatubo | 20.5 | 24 | 8 | 11 | 23.5 | 29 | 1 | 1 | 4 | 9 |
| Camiguin | 24 | 4 | 33 | 13 | 6 | 5 | 25.5 | 11 | 22.5 | 7 |
| Gamkonora | 18 | 9 | 5 | 8 | 23.5 | 9 | 25.5 | 27.5 | 22.5 | 8 |
| Karangetang | 11 | 12 | 15 | 17 | 23.5 | 15 | 9 | 13.5 | 22.5 | 26 |
| Kelut | 17 | 22 | 24 | 24 | 5 | 13 | 8 | 8 | 22.5 | 26 |
| Soputan | 2 | 10 | 3 | 4 | 23.5 | 29 | 25.5 | 27.5 | 22.5 | 26 |
| Dukono | 5 | 11 | 4 | 5 | 23.5 | 29 | 25.5 | 27.5 | 22.5 | 26 |
| Awu | 9 | 15 | 2 | 3 | 23.5 | 29 | 25.5 | 27.5 | 22.5 | 26 |
| Mayon | 25 | 21 | 21 | 19 | 23.5 | 29 | 5 | 4 | 22.5 | 26 |
| Gede-Pangrango | 15 | 14 | 14 | 15 | 23.5 | 17 | 25.5 | 27.5 | 22.5 | 26 |
| Semeru | 22 | 17 | 9 | 14 | 23.5 | 29 | 25.5 | 13.5 | 22.5 | 26 |
| Rinjani | 1 | 6 | 20 | 22 | 23.5 | 29 | 25.5 | 27.5 | 22.5 | 26 |
| Gamalama | 20.5 | 18 | 17 | 20 | 23.5 | 14 | 25.5 | 27.5 | 22.5 | 26 |
| Lamongan | 13.5 | 19 | 33 | 34 | 23.5 | 8 | 25.5 | 27.5 | 22.5 | 10.5 |
| Banda Api | 8 | 16 | 25 | 27 | 23.5 | 16 | 25.5 | 27.5 | 22.5 | 26 |
| Agung | 23 | 26.5 | 23 | 25 | 23.5 | 11 | 25.5 | 27.5 | 22.5 | 26 |
| Cereme | 19 | 25 | 18 | 21 | 23.5 | 29 | 25.5 | 27.5 | 22.5 | 26 |
| Tangkoko-Duasudara | 33 | 23 | 33 | 26 | 23.5 | 12 | 25.5 | 27.5 | 22.5 | 26 |
| Bulusan | 33 | 34 | 16 | 18 | 23.5 | 29 | 25.5 | 27.5 | 22.5 | 26 |
| Krakatau | 33 | 34 | 22 | 23 | 23.5 | 29 | 25.5 | 27.5 | 22.5 | 26 |
| Parker | 33 | 26.5 | 33 | 34 | 23.5 | 29 | 25.5 | 27.5 | 22.5 | 26 |
| Iya | 33 | 34 | 33 | 34 | 23.5 | 29 | 25.5 | 27.5 | 22.5 | 26 |
| Ranakah | 33 | 34 | 33 | 34 | 23.5 | 29 | 25.5 | 27.5 | 22.5 | 26 |
| Paluweh | 33 | 34 | 33 | 34 | 23.5 | 29 | 25.5 | 27.5 | 22.5 | 26 |
| Lewotolo | 33 | 34 | 33 | 34 | 23.5 | 29 | 25.5 | 27.5 | 22.5 | 26 |
| Lewotobi | 33 | 34 | 33 | 34 | 23.5 | 29 | 25.5 | 27.5 | 22.5 | 26 |
| Leroboleng | 33 | 34 | 33 | 34 | 23.5 | 29 | 25.5 | 27.5 | 22.5 | 26 |
| Iliwerung | 33 | 34 | 33 | 34 | 23.5 | 29 | 25.5 | 27.5 | 22.5 | 26 |
| Raung | 33 | 34 | 33 | 34 | 23.5 | 29 | 25.5 | 27.5 | 22.5 | 26 |
| Galunggung | 33 | 34 | 33 | 34 | 23.5 | 29 | 25.5 | 27.5 | 22.5 | 26 |
| Tambora | 33 | 34 | 33 | 34 | 23.5 | 29 | 25.5 | 27.5 | 22.5 | 26 |
| Taal | 33 | 34 | 33 | 34 | 23.5 | 29 | 25.5 | 27.5 | 22.5 | 26 |

**Figure 14: Individual rankings for calculated population exposure for the 50% dome collapse PDC hazard for all volcanoes across the five exposure categories. Exposure is provided for the smaller and larger volume scenarios using the 990 m buffer. Volcano order and cell colour as for Figure 12. Dome collapse PDCs are not VEI dependent and so the absolute ranks are not calculated, and these results are therefore applicable to the short-term estimate.**






| | Population (n) | | | | Buildings (n) | | | | Road length (km) | | | | Crop areas (km²) | | | | Urban areas (km²) | | | |
|---|---|---|---|---|---|---|---|---|---|---|---|---|---|---|---|---|---|---|---|---|
| **VEI** | 3 | 4 | 5 | A | 3 | 4 | 5 | A | 3 | 4 | 5 | A | 3 | 4 | 5 | A | 3 | 4 | 5 | A |
| Cereme | 3 | 2 | 1 | 11 | 2 | 1 | 1 | 8 | 2 | 2 | 2 | 12 | 1 | 1 | 1 | 6 | 1 | 1 | 1 | 7 |
| Merapi | 2 | 1 | 2 | 1 | 5 | 5 | 4 | 2 | 1 | 1 | 1 | 1 | 4 | 5 | 2 | 1 | 2 | 2 | 2 | 1 |
| Gede-Pangrango | 5 | 3 | 3 | 4 | 4 | 3 | 3 | 4 | 3 | 3 | 3 | 3 | 3 | 4 | 4 | 4 | 3 | 3 | 3 | 3 |
| Guntur | 4 | 5 | 4 | 5 | 3 | 2 | 2 | 3 | 6 | 6 | 6 | 7 | 5 | 3 | 3 | 5 | 5 | 5 | 5 | 6 |
| Mayon | 1 | 4 | 5 | 2 | 1 | 4 | 6 | 1 | 5 | 7 | 14 | 2 | 6 | 7 | 10 | 2 | 6 | 9 | 14 | 4 |
| Agung | 6 | 6 | 8 | 13 | 6 | 7 | 9 | 15 | 4 | 4 | 5 | 14 | 8 | 12 | 16 | 15 | 10 | 10 | 12 | 16 |
| Dieng Volcanic Complex | 9 | 7 | 7 | 9 | 7 | 8 | 8 | 9 | 8 | 10 | 7 | 11 | 10 | 11 | 13 | 10 | 11 | 13 | 10 | 11 |
| Semeru | 15 | 15 | 12 | 8 | 19 | 10 | 7 | 5 | 34 | 5 | 4 | 4 | 7 | 6 | 6 | 3 | 7 | 6 | 6 | 5 |
| Galunggung | 20 | 14 | 10 | 19 | 10 | 6 | 5 | 14 | 17 | 16 | 11 | 19 | 12 | 8 | 8 | 14 | 15 | 8 | 7 | 15 |
| Lokon-Empung | 8 | 8 | 6 | 6 | 11 | 9 | 10 | 10 | 9 | 9 | 10 | 9 | 30 | 31 | 30 | 27 | 8 | 7 | 8 | 10 |
| Sinabung | 16 | 18 | 20 | 27 | 13 | 13 | 12 | 24 | 12 | 15 | 15 | 24 | 2 | 2 | 5 | 13 | 18 | 19 | 17 | 27 |
| Papandayan | 10 | 9 | 11 | 17 | 18 | 16 | 14 | 21 | 10 | 13 | 18 | 20 | 14 | 14 | 15 | 17 | 12 | 14 | 13 | 17 |
| Lamongan | 11 | 10 | 9 | 7 | 26 | 24 | 22 | 16 | 7 | 8 | 8 | 5 | 16 | 17 | 12 | 11 | 9 | 12 | 11 | 8 |
| Kelut | 19 | 13 | 13 | 15 | 31 | 21 | 16 | 18 | 16 | 11 | 12 | 15 | 13 | 9 | 9 | 9 | 19 | 11 | 9 | 13.5 |
| Raung | 31 | 24 | 16 | 16 | 38 | 22 | 11 | 13 | 22 | 19 | 9 | 10 | 22 | 18 | 7 | 8 | 4 | 4 | 4 | 2 |
| Tengger Caldera | 25 | 19 | 18 | 12 | 8 | 12 | 15 | 6 | 18 | 12 | 17 | 6 | 17 | 13 | 14 | 7 | 24 | 20 | 19 | 13.5 |
| Soputan | 12 | 12 | 14 | 10 | 16 | 11 | 13 | 11 | 14 | 17 | 19 | 13 | 28 | 25 | 24 | 22 | 17 | 16 | 15 | 12 |
| Bulusan | 13 | 16 | 19 | 14 | 12 | 14 | 17 | 12 | 15 | 21 | 21 | 16 | 11 | 15 | 17 | 12 | 23 | 24 | 27 | 19.5 |
| Gamalama | 7 | 11 | 15 | 3 | 9 | 17 | 20 | 7 | 11 | 22 | 22 | 8 | 34 | 36.5 | 36.5 | 35 | 13 | 17 | 21 | 9 |
| Ranakah | 17 | 20 | 22 | 24 | 14 | 19 | 24 | 23 | 38 | 14 | 13 | 21 | 20 | 24 | 22 | 24 | 16 | 18 | 20 | 24 |
| Tangkoko-Duasudara | 21 | 23 | 26 | 29 | 15 | 15 | 18 | 22 | 13 | 18 | 20 | 25 | 34 | 33 | 31 | 35 | 14 | 15 | 16 | 18 |
| Iya | 14 | 17 | 21 | 21 | 20 | 18 | 19 | 20 | 30 | 30 | 32 | 31 | 31 | 32 | 34 | 35 | 20 | 21 | 24 | 24 |
| Rinjani | 29 | 21 | 17 | 20 | 32 | 28 | 23 | 26 | 35 | 20 | 16 | 18 | 26 | 23 | 19 | 20 | 36 | 23 | 18 | 24 |
| Camiguin | 18 | 22 | 25 | 26 | 21 | 23 | 27 | 28 | 19 | 24.5 | 28 | 29 | 23 | 26 | 27 | 27 | 25 | 31 | 31 | 30.5 |
| Lewotobi | 22 | 26 | 27 | 23 | 28 | 31 | 32 | 27 | 23 | 28 | 27 | 22 | 19 | 20 | 23 | 19 | 21 | 25 | 26 | 19.5 |
| Taal | 32 | 28 | 23 | 22 | 23 | 25 | 25 | 19 | 33 | 34 | 24 | 26 | 18 | 19 | 21 | 18 | 32 | 29 | 23 | 24 |
| Awu | 23 | 27 | 29 | 25 | 17 | 20 | 26 | 17 | 21 | 24.5 | 26 | 23 | 34 | 36.5 | 36.5 | 35 | 22 | 22 | 25 | 24 |
| Lewotolo | 28 | 29 | 31 | 30 | 25 | 27 | 30 | 29 | 20 | 23 | 31 | 27 | 21 | 22 | 26 | 23 | 28 | 26 | 30 | 28.5 |
| Suoh | 38 | 39 | 40 | 39 | 33 | 35 | 37 | 38 | 27 | 26 | 25 | 34 | 15 | 16 | 18 | 21 | 27 | 27 | 29 | 34 |
| Pinatubo | 40 | 40 | 39 | 40 | 22 | 26 | 29 | 32 | 38 | 39 | 39 | 39 | 9 | 10 | 11 | 16 | 33 | 34 | 37 | 38.5 |
| Parker | 35 | 33 | 28 | 38 | 40 | 37 | 21 | 39 | 26 | 27 | 23 | 37 | 24 | 21 | 20 | 27 | 39 | 39 | 36 | 38.5 |
| Leroboleng | 30 | 32 | 34 | 34 | 35 | 36 | 36 | 35 | 24 | 29 | 30 | 32 | 27 | 29 | 29 | 35 | 26 | 28 | 28 | 30.5 |
| Gamkonora | 24 | 25 | 30 | 28 | 30 | 33 | 33 | 31 | 29 | 32 | 36 | 30 | 38 | 39 | 39 | 35 | 29 | 30 | 32 | 34 |
| Karangetang | 26 | 30 | 32 | 18 | 34 | 34 | 36 | 25 | 25 | 31 | 34 | 17 | 32 | 34.5 | 35 | 35 | 30 | 32 | 35 | 21 |
| Dukono | 34 | 35 | 24 | 33 | 29 | 30 | 28 | 33 | 38 | 36 | 29 | 36 | 38 | 34.5 | 33 | 35 | 39 | 36 | 22 | 34 |
| Iliwerung | 36 | 38 | 36 | 36 | 37 | 38 | 38 | 35 | 28 | 33 | 33 | 28 | 25 | 28 | 28 | 27 | 35 | 35 | 33 | 34 |
| Tambora | 39 | 34 | 33 | 37 | 24 | 29 | 31 | 34 | 38 | 37 | 35 | 38 | 38 | 27 | 25 | 27 | 37 | 38 | 39 | 38.5 |
| Paluweh | 27 | 31 | 35 | 32 | 36 | 39 | 39 | 37 | 32 | 38 | 38 | 35 | 29 | 30 | 32 | 35 | 34 | 37 | 38 | 34 |
| Banda Api | 33 | 36 | 37 | 35 | 27 | 32 | 34 | 30 | 31 | 35 | 37 | 33 | 38 | 39 | 39 | 35 | 31 | 33 | 34 | 34 |
| Krakatau | 37 | 37 | 38 | 31 | 39 | 40 | 40 | 40 | 38 | 40 | 40 | 40 | 38 | 39 | 39 | 35 | 39 | 40 | 40 | 38.5 |

**Figure 15: Individual rankings for calculated exposure using the 50% column collapse PDC hazard. Column names, volcano order and cell colour as for Figure 12.**

Overall, the consideration of eruption frequency into the rankings does not considerably change the overall trend across our volcanoes, scenarios, hazards or exposure categories (Supplementary Material 3). Disparity in rankings across the volcanoes is strongly driven by variability in location affected, and thus exposure. There are nevertheless interesting case-studies to be observed. Considering the population exposure to 1 kg/m² of tephra fall (Figure 12), Cereme ranks 1st when considering the conditional occurrence of eruptions of VEI≥4, but ranks 12th when a long-term hazard assessment is

considered. In contrast, Raung volcano ranks between 11–18 when considering the conditional occurrence of VEI 3–5, but

ranks 4th when considering a long-term approach. These different behaviours lie in the eruptive histories of the individual volcanoes and the computation of the probabilities of occurrences of each VEI (Section 2.3). Although the occurrence of large eruptions at Cereme results in high exposure (Figure 6), eruptions of VEI 3, 4 and 5 have annual probabilities of occurrence of 0.4%, 0.2% and 0.1%, respectively. By contrast, simulated eruptions from Raung result in, on average, one third of the total exposure attributed to Cereme, but their annual probabilities are on average one order of magnitude higher (e.g. 3.6%, 1.6% and 0.6% for VEI 3, 4 and 5, respectively). This observation highlights the benefits and pitfalls of short-vs long-term hazard assessments, and their combined use and understanding is required to fully inform decision-making during various phases of volcanic crises.

## 5 Limitations

As with any consistently applied regional approach to hazard or exposure assessment, there are limitations to using widely available data. We discuss these limitations in more detail over the next sections to highlight how our results may differ with further data and/or study.

### 5.1 Hazard approach

A regional approach to hazard simulation can omit local context (e.g. recent unrest crises) and data (e.g. unpublished eruption records) that would be included within a volcano-specific hazard assessment. By employing more generic inputs across all volcanoes, our results are relevant and comparable at a regional scale, but caution should be used in considering such assessments at the individual volcano scale. However, they do provide a solid foundation from which more detailed assessments can be applied. Specifically, the following factors could be improved in a local single volcano assessment:

★ By using global datasets for ESPs (e.g. GVP, VEI classification), datasets can be biased towards particular eruptions, and more recent times.

★ Since we modelled hazard probabilistically across 40 volcanoes, we were constrained to using empirical models that do not fully capture the physical processes underpinning volcanic phenomena. This is an unavoidable consequence of the computational power required for physical models.

★ For tephra, we considered the hazard from both tephra fall and large clasts, and for PDC, we considered the hazard from both column collapse and dome collapse generation mechanisms. Not all volcanoes are likely to produce all hazard types, and we do not distinguish here; therefore overall rankings, i.e. the ordering of volcanoes in Figures 12-15, may require further interpretation for certain volcanoes. However, individual values and rankings are still appropriate and we provide all data so that the reader can choose certain assessments only if preferred.





★ Some of our case-study volcanoes have produced PDCs that differ in their generation mechanism, and thus dynamics,
from the dome and column collapse mechanisms simulated here. For example, the 2010 eruption of Merapi produced
PDCs from boiling over, dome explosion/lateral blast, fountain collapse and dome collapse over the course of 11 days
(Komorowski et al., 2013; Jenkins et al., 2016). In the case of repeated PDCs, our modelling does not capture
modification of the subsurface topography or smoothness as a result of previous deposits, which would affect runout.
Additionally, the use of the SRTM 2000 DEM for modelling could result in less reliable inundation areas where major
topographic changes have occurred since its acquisition, although we did not observe this effect at Merapi (Figure 4
a,b).

★ We did not include lahars in this analysis because: i) the occurrence of lahar is dependent upon the presence of
unconsolidated deposits produced by a prior eruption, ii) lahars can initiate from previous lahar deposits so their impact
over time and space is hard to capture without detailed volcano-specific study, iii) localised variations in rainfall can
strongly influence the probability of lahar occurrence and iv) empirical models that enable large numbers of
simulations, like LAHARZ, do not capture debris flow and hyperconcentrated flows that are typical of this region
(Iverson et al., 1998; Lavigne and Thouret, 2003).

★ To constrain the scope of this study, other volcanic hazards such as lava, gases, volcanogenic tsunami and lightning
were not included.

## 5.2 Exposure data

The limitations and features of regionally applicable exposure data have been well detailed for our data sources (see
references in Section 2.4), although the interpolation or extrapolation of our data to a consistent grid for calculation across
different hazards and exposure categories inevitably meant that some resolution in data was lost. For example, we
disaggregated the number of buildings and people within a grid cell and calculated exposure to a hazard as the proportion
of each of our 90 x 90 m cells covered by the hazard so that any clustering of buildings at the original scale (~1 km$^2$ for
people and 36 m$^2$ for buildings) has been lost; we don't expect this to have a major effect on our overall results but for
detailed local inspection there may be some variation as a result. We also noticed a small number of irregularities in our
building exposure results that unavoidably arose as a result of the dataset limitations, and we note them here; as with the
interpolation, they do not have a major effect on results but would be worth investigating further if results are interpreted
at the individual volcano scale:

★ The GHSL data used to spatially distribute buildings exhibits a 300 km long horizontal line through central Java that
appears to overestimate built up areas immediately to the south and underestimate built up areas immediately to the



north. This affects the distribution of our buildings and the artifact comes within 30 km of several volcanoes in our analysis.

★ A second artefact in the GHSL is its interpretation of built-up areas using remote-sensing. We found that in a small number of specific locations, bare rock areas such as riverbeds (e.g. to the northwest of Kelud) or volcanic craters (e.g. Gede-Pangrango) had been misinterpreted as built-up areas resulting in the placement of buildings into areas where they are unlikely to exist.

## 6 Discussion and conclusions

With this study we have evaluated multiple categories of exposure (n=5) to a range of volcanic hazards (n=4) and VEI scenarios (n=3) to give probabilistic outputs for 40 high-threat volcanoes. Ranking was performed using both a "short-term" approach, where the exposure is conditional on the occurrence of the eruption scenario, and a "long-term" approach, which accounts for the probability of occurrence of a given eruption scenario at each volcano. We explicitly list our simplifications and how different initial conditions were determined. This work expands significantly upon previous
approaches to regional volcanic hazard and exposure assessment that considered concentric radii to reflect hazard extent and/or population exposure only. By probabilistically modelling multiple volcanic hazards and coupling them with open-access exposure data, our approach provides a consistent and transferable method for comparing hazard and exposure at a volcano and across multiple volcanoes, hazards and exposures. While the modelling provides valuable information that can act as a foundation to more detailed local assessments, especially for volcanoes that have limited or no hazard and
exposure assessments already conducted, it is not intended to replace local assessments. Wherever possible, local context, data and knowledge should all be incorporated.

We found Merapi to pose the greatest threat when all hazards, exposures and VEI scenarios are considered with equal weighting. For a VEI 4 scenario, a c. 1 in 100 year event at Merapi, approximately 7.8 million people, 210,000 buildings, 38,000 km of road, 930 km$^2$ of crops and 1,150 km$^2$ of urban area have a 50% probability of being affected by tephra fall
accumulations $\geq 1$ kg/m$^2$. The threat that Merapi poses is well appreciated and it is likely one of the most studied volcanoes in Indonesia. A key aim of our study was to highlight those volcanoes that may have been overlooked, perhaps because they are not frequently or recently active, but that have the potential to affect large numbers of people and assets. Guntur volcano in Java fits that description well, with comparable, and in some cases larger (e.g. Figure 8) exposure than Merapi. Retnowati et al. (2018b) carried out a current and projected exposure estimate for concentric radii of lava flow and
exponentially thinning tephra fall, extending it towards estimates of building damage and loss. A more detailed local hazard





and risk assessment for Guntur would be of high value, especially as the volcano appears to be a closed system at present so that a future eruption may be larger than those experienced in the recent past.

The GIS framework developed for this work is modular with the code freely provided (github.com/vharg/VolcGIS) so that future works can simply plug in updated or improved hazard or exposure data. For example, the key improvements that we
anticipate will be most influential in improving our findings are:

★ Field studies: improving our knowledge of the past behaviour at volcanoes in the region, and their likely ESPs, will help us refine our model outputs. The rankings provided by this method can support the prioritisation of which volcanoes to focus risk reduction activities on.

★ The incorporation of more sophisticated hazard models that can better describe the physical processes underpinning
volcanic hazards; such models also require greater data and computing resources, which will hopefully improve over time.

★ The open-access data underpinning our hazard and exposure assessment, e.g. DEMs, Open Street Map, are expected to improve in quality and resolution going forward and these can be used within the framework to provide updated, higher resolution outputs.

★ A robust and evidence-based method for aggregating exposure scores across multiple hazard and exposure categories, potentially multiple different aggregations are needed to cover diverse aspects such as life safety, loss of livelihoods or economic impacts.

We also identify a number of further areas for study that could widen the assessment provided here:

★ This study is limited to the quantification of the exposure of populations and their assets to a range of volcanic hazards.
Future efforts should contribute to the development of applicable - rather than theoretical - models to quantify critical aspects of vulnerability which, when incorporated into such GIS frameworks as the one proposed here, will allow to estimate measures of impact and risk as a function of the spatial distribution of hazard intensities and exposed assets.

★ Efforts to better constrain the relationship between hazard intensity and impact have dominantly focused on the hazard caused by tephra fallout. In parallel, the impact of other hazards is often oversimplified. For instance, our method
considers a binary impact from PDCs where inundation implies impact. Recent studies have demonstrated that this assumption is disproved by field observations (Jenkins et al., 2013; Lerner et al., under review). Shifting from estimating exposure to impact for flows requires advances in two directions. Firstly, there is a need for new flow models that predict not only a binary inundation but also some measure of impact intensity metrics (e.g. flow depth, dynamic pressure) whilst requiring ESPs that can realistically be estimated for purposes of hazard assessments. We





acknowledge that the complexity of the physical processes governing such flows makes this task challenging. Secondly, more research should be dedicated to investigate how, when and why flows can affect populations and their assets. Again, the diversity of flows (e.g. dense vs dilute components for PDC) makes this task complex, but post-event impact assessments, experimental and theoretical studies all contribute to establishing the baseline for better vulnerability and impact models.

★ Finally, volcanic risk is intrinsically *dynamic*. On the one hand, hazards can interact in a nonlinear fashion. For instance, forecasting lahar triggering is challenging as it depends on the properties of the fresh pyroclastic deposit, the topography and the rainfall magnitude and intensity. Similarly, large clasts can perforate roofs, but the presence of tephra might cushion the impact and reduce this hazard (Williams et al., 2019). On the other hand, exposure and vulnerability also vary in space and time. For instance, the risk to the tourist hikers in Southeast Asia varies as a

function of the day and the season, exposing populations from various cultures and awareness of volcanic hazards. Here, we have explored the variability of population exposure as a function of hazard seasonality, and the proposed framework could also be applied to estimate the changes in exposure using yearly datasets of land cover and population. Future efforts should therefore aim at modelling the impact and risk from volcanic eruptions as a dynamic rather than static process



**Appendix A: Model input parameters**

**Tephra fall**

| Hazard; Model | Parameter | Inputs | | | Data source/Rationale |
|---|---|---|---|---|---|
| | | *VEI 3* | *VEI 4* | *VEI 5* | |
| Tephra fall; Tephra2 (Bonadonna et al. 2005) | Erupted mass (kg) | 3.2 x 10^10 | 3.2 x 10^11 | 3.2 x 10^12 | The midpoint of the logarithmically bounded range of bulk volume provided by the VEI classification, assuming a bulk density of 1000 kg/m³. |
| | Plume height (km above vent) | 13 | 20 | 27 | Based on the original classification of Newhall and Self (1982). Single values, rather than stochastically sampled ranges, were used to prevent the simulation of a broad spectrum of eruption intensities that could not be equally applied across the wide range of volcanoes and eruptive styles considered in this regional study, thus making results non-comparable across volcanoes. The plume height is the mid-point of the calculated column heights using Eq. 3 of Mastin et al., (2009) based on the minimum and maximum volumes defining each VEI. |
| | TGSD: mean ($\mu$) S.D. ($\sigma$) (phi) Grain size range: 7 to -6 phi | $\mu = -0.74$ $\sigma = 2.4$ | $\mu = 0.9$ $\sigma = 1$ | $\mu = 1.35$ $\sigma = 1.16$ | The total grain size distribution (TGSD) is one of the most difficult parameters to constrain since it is dependent on the collection of well preserved field data (Pioli et al., 2019). We use analogue TGSDs of Ruapehu (1996; Bonadonna et al 2005) for VEI 3, Kelud (2014; constrained from TEPHRA2 inversion modeling by Williams et al. 2020) for VEI 4 and Pinatubo (1991) for VEI 5 (Volentik; 2009 compiled from Koyaguchi and Ohno; 2001). |
| | Particle density (kg/m³) | Pumice: 1000 Lithics: 2600 | | | Scollo et al. (2008) suggested that tephra density does not greatly affect the simulated results in Tephra2 and so we use typical values here. |
| | Plume model | alpha = 3; beta = 1.5 | | | Uses a beta distribution to constrain a plume with the majority of tephra dispersed at ~80% height. |
| | Diffusion coefficient (m/s) | 5000 | | | These empirical parameters describing atmospheric diffusion in Tephra2 should ideally be constrained through inversion of field deposits. As this is not possible here, for consistency across the regional analysis, we use the values of Biass and Bonadonna (2012) for subplinian/Plinian eruptions of Cotopaxi volcano. |
| | Fall time threshold (s) | 4000 | | | |
| | Eddy constant (m²/s) | 0.04 | | | Standard for the Earth's atmosphere. |
| | Wind conditions | Synoptic hourly data from a 10 year record (2010-2019) at the point closest to each volcano. Geopotential height, u- and v-wind components were retrieved at a spatial resolution of 0.25° for 37 pressure levels from the European Center for Medium-Range Weather Forecasts (ECMWF) ERA5 (Hersbach et al., 2020) - the highest temporal and spatial reanalysis dataset available. The data was formatted to single profiles for Tephra2, resulting in 2,880 profiles per volcano (10 years × 12 months × 24 hours). | | | |

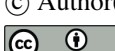



**Large clast**

| Hazard; Model | Parameter | Inputs | | | Data source/Rationale |
|---|---|---|---|---|---|
| | | *VEI 3* | *VEI 4* | *VEI 5* | |
| Large clast; Rossi et al (2019) | Clast density (g/cm³) | 2.5 | | | Lithic size corresponding to a kinetic energy of 30 J, identified as a threshold for skull fracturation and roof penetration. A similar energy can be produced by a 5.6 cm pumice with a density of 0.63 g/cm³ |
| | Clast diameter (cm) | 3 | | | |
| | Atmospheric conditions | As for tephra fall, with the additional parameters of temperature and humidity, the latter being used to estimate air density and viscosity. Three-dimensional atmospheric data were retrieved using the *LagTrack* code (Poulidis et al., 2021). | | | |
| | Topography | Elevation data source the Shuttle Radar Topography Mission (SRTM) 1 Arc-Second Global dataset. acquired in 2000 to provide a continuous elevation surface at a resolution of ~30 m (Farr et al., 2007). The extent of each selected volcano is set to be 60 km from the vent. The higher resolution ~8 m DEMNAS (Julzarika and Harintaka, 2019) was found to be less accurate for steep volcanic terrains, and is only available for Indonesia. | | | |

**Dome collapse PDCs**

| Hazard; Model | Parameter | Inputs | | Data source/Rationale |
|---|---|---|---|---|
| | | *Small volume scenario* | *Large volume scenario* | |
| Dome collapse PDC; LAHARZ (Iverson et al., 1998) developed by Schilling (1998) and adapted to MATLAB by Rudiger Escobar-Wolf, using the PDC calibration of Widiwijayanti et al. (2009) | Volume (m³) | 4.5 x 10⁵ | 9.8 x 10⁶ | This semi-empirical model is based on a scaling argument that relates flow volume (V) to channel cross sectional area (A) and planimetric area (B) as follows: A = CV^2/3, B = cV^2/3. The model was originally calibrated using data from 27 lahars (C = 0.05, c = 200), however more recent calibrations have been undertaken to derive coefficients for alternative flow types, including PDCs (Widiwijayanti et al., 2009). In this work we use the calibration presented by Widiwijayanti et al., (2009) (with C = 0.05, c = 40), which is based on data for BAF's acquired at Soufriere Hills, Merapi, Colima and Unzen volcanoes. Simulated flow volumes are the 50th and 90th percentiles obtained from the global block and ash flow dataset Flowdat (Ogburn, 2012). The 10th percentile is not included here as such volume usually results in flows restricted to the crater area. |
| | Topography | As for large clasts | | |



**Column collapse PDCs**

| Hazard; Model | Parameter | Inputs | | | Data source/Rationale |
|---|---|---|---|---|---|
| | | *VEI 3* | *VEI 4* | *VEI 5* | |
| Column collapse PDC; ECMapProb (Araveno et al., 2020) | Column collapse height (m) [Variability applied] | 1300 [130] | 2000 [200] | 2700 [270] | The height of column collapse for sustained eruptions has been suggested to represent ~10% of the total column height (Wilson et al. 1978) and so we consider our collapse heights as 10% of the heights used for tephra fall modeling. A +/- 10% range was applied to represent the variability in this assumption. |
| | Vent location [Variability applied] | Summit or centre of active crater [crater radius] | | | The vent location was selected based on the summit or active crater centre of each volcano, determined from Google Earth and eruption records. Variability was based on the size of the summit area or crater at each volcano, determined from Google Earth. |
| | H/L [Variability applied] | 0.24 [0.08] | | | The H/L ratio (a value based on the ratio of the height to length travelled by flows in the past) was taken from the median value in pumice flow category of the FlowDat database (Ogburn, 2012). The variability represents the middle 50% of values in the FlowDat database (Ogburn, 2012). |
| | Topography | As for large clasts and dome collapse PDCs | | | |


**Appendix B: Probabilistic forecasting of dome collapse PDC travel directions**

A MATLAB implemented methodology was developed to rapidly analyse a volcano's summit topography, using this to assign probabilities to the travel directions of future effusive flows (here applied to dome collapse PDCs) . Inputs to the code include: a DEM, coordinates of the crater center, the radius of the crater or summit region and the swath length. The swath length is the entire length from the start point over which topography is considered in the calculation; it should extend outside of the crater or summit region and include any localized topographic highs. For this study we have used the 'summit width' parameter obtained from the global database of composite volcano morphology (Grosse et al., 2014) to identify swath lengths, with the addition of a 20% buffer to ensure the full summit topography was included in the calculation. For volcanoes not present in the database, a default length of 1500 m is used.  The procedure is as follows:

1. Upon acquisition of the swath length, 360 swath profiles *(SW$_{1:360}$)* (each with a width of 50 m) are created radiating from the starting coordinate to the full swath length. Each swath consists of *n* cells from start to the full length (see Figure A2.1)

2. At each cell along the length of the swath, elevation values *(E)* are compared with elevation values in the total population of swaths at position *n* by computing percentiles *(P)*. In Figure B2.1 cell populations are denoted by colour, for example all $E_1$ values are considered a population, as are all $E_2$ ...$E_n$. This way elevation is analysed at each radial distance step from the start point to the full swath length. Elevation values $E_{1:n}$ are transformed into $P_{1:n}$ such that (Eq. 1):

    $SW_1(E_1), SW_2(E_1) ...SW_{360}(E_1) \rightarrow SW_1(P_1), SW_2(P_1) ...SW_{360}(P_1)$
    $SW_1(E_2), SW_2(E_2) ...SW_{360}(E_2) \rightarrow SW_1(P_2), SW_2(P_2) ...SW_{360}(P_2)$
    ...
    $SW_1(E_n), SW_2(E_n) ...SW_{360}(E_n) \rightarrow SW_1(P_n), SW_2(P_n) ...SW_{360}(P_n).$

    (1)

3. Percentiles are summed down-swath to get a final *(V)* value that can be considered a proxy for the elevation. Values are inverted and interpolated to 10° intervals, such that high *(V)* values are the more likely flow directions (Eq. 2):

    $V_i = \Sigma \, SW_i(P_{1:n})$

    (2)

4. To estimate probabilities (Pr) for each swath, we calculate (Eq. 3):

5. $Pr_i = V_i / \Sigma \, V_{1:36}$

    (3)




The tool outputs a table featuring 36 10°, azimuth bins, their associated probabilities and the XY coordinates for each start point. In this work output coordinates were fed into the dome collapse PDC-calibrated LAHARZ, and dome collapse PDCs were simulated in all directions. Binary LAHARZ output hazard footprints were multiplied by their travel direction

probability and aggregated to produce a final conditional dome collapse PDC probability raster that quantifies both the probability of travel direction and the probability of inundation at each grid cell.

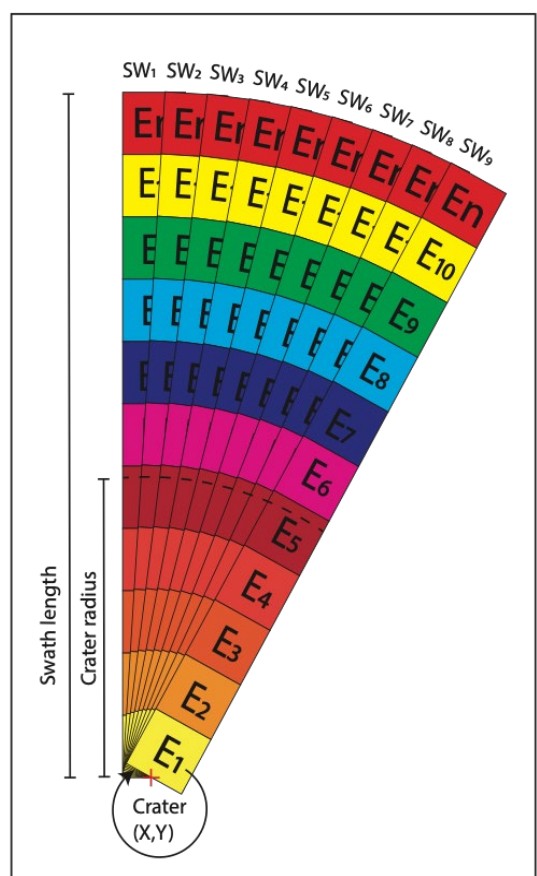

**Figure B1: A graphical representation of the methodology used to quantify dome collapse PDC travel direction probabilities. Radial swaths (SW$_i$) are initiated from the crater start point (X,Y). For each distance step from the start to the full swath length, populations of elevation values E$_i$ (represented by cells of the same colour) are compared and the percentile calculated.**

**Code availability**

The open-source Python code, *VolcGIS,* which implements all of the spatial operations required for our exposure analyses is freely available at github.com/vharg/VolcGIS.

**Data availability**

All hazard and exposure data, and associated format descriptions, are provided in user-friendly format in the Supplementary Material, and are available via the links provided (only available to reviewers at this stage).

*Supplementary Material SM1: Hazard model outputs*

*Supplementary Material SM2: Eruption Frequency-Magnitude*

*Supplementary Material SM3: Exposure results*

**Author contributions**

**SFJ** and **SB** conceived the project, co-ordinated group activities, and oversaw hazard modelling. **SFJ** processed and analysed wind data and drafted the final manuscript. **SB** modelled large clast hazard, developed and wrote the GIS framework and performed the exposure analysis. **GTW** carried out the building exposure modelling. **JLH** carried out the road exposure modelling and calculated frequency-magnitude relationships. **EMT** developed the dome collapse PDC travel

directions methodology (Appendix B) and modelled dome collapse PDCs. **QY** modelled tephra fall hazard. **VB** performed the analysis for volcano selection. **ESM** carried out the population exposure analysis, calculating PEI. **GAL** modelled column collapse PDCs. **MS** retrieved and pre-processed all DEMs. **AV** developed key figures. All authors contributed to concept development, writing, reviewing and editing the manuscript.

**Competing interests**

The authors declare that they have no competing interests.

**Disclaimer**

The information set out in this publication reflect the author's views.



**Acknowledgements**

We would like to thank Eduardo Rossi for his support in implementing the large clast model, Álvaro Aravena Ponce for

his support with ECMapProb, and Rudiger Escobar-Wolf for providing the Matlab implementation of LaharZ. We are also

indebted to Edwin Tan, for his unending support in the use of the ASE/EOS High-Performance Computing Cluster, Gekko.

This research was supported by the Earth Observatory of Singapore via its funding from the National Research Foundation

Singapore and the Singapore Ministry of Education under the Research Centres of Excellence initiative and comprises

EOS contribution number 412. Support was provided to SFJ and JLH by the AXA Research Fund as part of a Joint Research

Initiative on Volcanic Risk in Asia, and to SFJ, QY and GAL by Singapore's National Research Foundation (Project

Number: NRF2018NRF-NSFC003ES-010). The globally disruptive driver of this study was COVID-19 and the resulting

lockdown/s, which spurred the need to come together (remotely) and work on a common research project.

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
