# Peer review of "Evaluating and ranking Southeast Asia's exposure to explosive volcanic hazards"

_Natural Hazards and Earth System Sciences, 2021_

## Author Comment (AC1)

**Response to Reviewer 1**

1. **I see that the threshold used for marking the onset of damages of the buildings, related to ash fall, was set to 100 kg/m². As noticed by the authors this is true for weak buildings (line 222). Is there an estimate of the mean typology of the buildings in the considered areas or are you using a ``conservative'' approach?**

There are few detailed estimates of fragility for roofs in the region, and only one that considers the different degrees of damage that might be sustained by tephra fall loading on roofs typical to the area (Williams et al., 2020). This supports our choice of tephra load threshold as a median estimate for damage and we now include reference to this study in our text, as below:

*Revised text:* "Based on remote damage surveys around Kelud volcano, Java, Williams et al. (2020) identified 100 kg/m² as the median tephra load associated with moderate or worse damage to tiled or metal sheet roofs: roof types that are common across Indonesia and the Philippines."

2. **The nomenclature used for the concepts of ``short-term'' and ``long-term'' hazard seems different from what is found in the literature. The authors (lines 93-96 and 523-525) associate ``short-term'' with the hazard conditional to a given scenario and ``long-term'' with the probability of occurrence of the eruption scenario.**
   **However, other authors (eg. Marzocchi and Bebbington, 2012) associate the terms ``short'' and ``long'' strictly with the time scale and use ``short-term'' to indicate a forecast in a time horizon of hours/weeks or months, typically of interest in managing evolving episodes of volcanic unrest; and ``long-term'' for time windows of years to decades that are required for land use and evacuation planning.**
   **This point should be clarified and, in case of conflict with the commonly used nomenclature, I suggest to modify the terms or explicitly state that you are using ``short-term'' and ``long-term'' in a more specific way.**

We have revised our terminology around the use of short-term and long-term, and reframed our analysis to not refer to timeframes in order to address this very valid point. We now reference "conditional probabilities" (previously "short-term") to reflect when the assessment was conditional upon the considered eruption scenario occurring at that volcano, and "absolute probabilities" (previously "long-term") to reflect when the assessment incorporated the probability of the eruption scenario occurring.

3. **Concerning ``the need for new flow models that predict not only a binary inundation but also some measure of impact intensity metrics (e.g. flow depth, dynamic pressure'' (line 717), in generally I agree with the authors. However, some models are already freely available (see eg: https://github.com/TITAN2D/titan2d or https://github.com/demichie/IMEX\_SfloW2D)**

Very true, thank-you for highlighting this oversight. We omitted reference to probabilistic simulation, which makes the use of the 2D flow models that do incorporate flow dynamics challenging (but not impossible) because of runtimes and storage sizes. We have reworded our text to recognise this:

*Revised text:* "Shifting from probabilistically estimating exposure to impact for flows requires advances in two directions. Firstly, there is a need for flow models compatible with probabilistic approaches that predict not only a binary inundation but also some measure of impact intensity metrics (e.g. flow depth, dynamic pressure) whilst requiring ESPs that can realistically be estimated for purposes of hazard assessments."

**References**

Williams, G.T., Jenkins, S.F., Biass, S., Wibowo, H.E. and Harijoko, A., 2020. Remotely assessing tephra fall building damage and vulnerability: Kelud Volcano, Indonesia. *Journal of Applied Volcanology*, *9*(1), pp.1-18.

---

## Author Comment (AC2)

**Response to Reviewer 2**

1. **Lahars: in the present version (if I am not wrong) the reader needs to read down to the end of the "Limitations" section (Section 5) to discover why they are not considered in the present manuscript. I myself, as a reader, have been wondering why lahars were not taken into account in assessing volcanic hazard from Southeast Asian volcanoes (while, for example, large clasts' impact is) until I got to that section in the end of the paper. However, this is a strong limitation that should be put forward from the beginning, in my opinion. I acknowledge that it would be an extra effort to account for lahar hazard in a similar way as done for tephra load or large clasts' impact, and I am not asking to do this. I also agree with what the author state in line 645, i.e. that the available empirical models are not able to simulate some of the known flow types, and more in general are not able to capture the details of lahar propagation. However, probably the same can be said for the energy cone model, that still is used in this study, and I agree on using it in a "hazard perspective", as I am one of the authors of a cited paper (Tierz et al, 2016) in which we showed that, notwithstanding its oversimplifications, the energy cone model is able to statistically reproduce the recorded PDC-invaded areas from Vesuvius and Campi flegrei eruptions.
So I would advice to add a paragraph already in the Introduction section on why only the 4 hazardous events are considered, and not lahars, for example. If possible, it would be interesting to see whether lahar hazard and exposure could be taken into account at a few volcanoes (maybe volcanoes characterized by very different lahar hazard), and see whether the introduction of lahars would significantly change their relative ranking (a kind of sensitivity analysis on the hazard events considered). I realize this is probably a large extra work, so I leave it just as a suggestion.**

We like the idea of a sensitivity test, and when we were developing this work we did spend a long time trying to simulate lahars on our test volcanoes (using LaharZ). However, we could not consistently recreate historical lahars with the DEMs and source parameters available to us. Mostly this was because the empirical coefficients for LaharZ are not calibrated for southeast Asian volcanoes, but even when we modified the coefficients, the dynamic and deposit-dependent nature of lahars meant that we did not feel confident incorporating lahars into our assessment. For example, where prior PDCs had deposited material >5 km (i.e. outside of the energy cone-river intersection we were using to define a starting point) from the volcano, the runout was much underestimated in our modelling (e.g. Merapi 2010, Gendol channel: footprints from de Bélizal et al., 2013). Linking lahar start point and volumes with likely distribution of pyroclastic material from PDCs and tephra is needed, something we felt required a much larger and more specific investigation that was outside the scope of this study. We therefore decided not to include lahars in our assessment. This is also the reason we do not want to include a section on incorporating lahars as a sensitivity test, even though we recognise that rain-triggered lahars especially are a major hazard in southeast Asia.

We have moved the bulletpoint from the Limitations to the Introduction and added some further explanation for why we did not consider lahars, as follows:

*Revised text:* We recognise that rain-triggered, and occasionally lake breakout, lahars are key hazards in Indonesia and the Philippines (Lavigne et al., 2007; Newhall and Punongbayan, 1996). However, they are not included in our assessment because i) their runout and inundation area is directly controlled by the spatial distribution and characteristics of previously emplaced pyroclastic material; ii) they can be produced independent of an eruption so that their impact over time and space is hard to capture without detailed volcano-specific study; iii) localised

variations in rainfall can strongly influence the probability of lahar occurrence; and iv) empirical models that enable large numbers of simulations, like LAHARZ, have not been calibrated for lahars in southeast Asia and do not capture the dynamics of the debris and hyperconcentrated flows typical of this region (Iverson et al., 1998; Lavigne and Thouret, 2003).

2. **Similar considerations apply to the conclusion that Krakatau is the last in the ranking… The reason is probably that tsunamigenic explosions have not been considered in the hazard and exposure analyses, and this affects the ranking. So I think this should be mentioned.**

Agreed, we have included text referencing the low ranking for Krakatau, when the four hazards we simulate are considered, and agree that the inclusion of volcanogenic tsunami would alter the ranking.

*Revised text:*

[Start of Section 4: Volcano ranking] The multi-hazard and multi-exposure analysis presented here allows us to rank all 40 volcanoes according to their exposure to the four volcanic hazards simulated here.

[Prior to Figure 12]: As we did not simulate all volcanic hazards, volcanoes at the lower end of the ranking cannot be assumed to pose low threat from all volcanic hazards. For example, Krakatau volcano ranks as our lowest threat volcano (Figure 11) but the 2018 tsunamigenic flank collapse, which killed more than 400 people (Williams et al., 2019), highlights the importance of considering other volcanic hazards and conducting volcano-specific field studies to determine a volcano's overall threat.

3. **Another point not clear to me is what probabilities have been used to evaluate the VEI scenarios when they are considered separately in the "short-term" evaluation. In general, I would suggest:**

   - **throughout the manuscript, avoid using "short-term" and "long-term" to distinguish between the two cases illustrated, as they are effectively both "long-term" assessments: for the tephra fallout in particular this is misleading, as in a real short-term assessment one would never use the yearly or monthly statistics of the wind to assess tephra fallout hazard, but would use the current weather forecast for the upcoming days. My suggestion is to use "conditional to an eruption size" and "absolute" respectively for what is now called "short-term" and "long-term", because this is what they are (as also called by the authors in lines 147-148).**

We have revised our terminology around the use of short-term and long-term, and reframed our analysis to not refer to timeframes in order to address this very valid point. As suggested and already used in our manuscript, we now reference "conditional probabilities" (previously "short-term") to reflect when the assessment was conditional upon the considered eruption scenario occurring at that volcano, and "absolute probabilities" (previously "long-term") to reflect when the assessment incorporated the probability of the eruption scenario occurring.

   - **rewrite section 2.3: I did not understand how the frequency magnitude relationships are calculated (a paper in prep is cited, so I could not get to know more). Also I assume that, when the "short-term" evaluations are computed, no probability of the VEI is used to weight hazard and exposure (correct?), while when all the VEIs together are considered ("long-term" or absolute case) the annual probabilities of the different VEIs are used to weight the hazard and exposure**

**(according to the total probability theorem). However, lines 311-315 are quite confusing: from those, apparently also for the separate VEI scenarios (short-term case) a weight is given, but I cannot understand why. This should be made much clearer.**

Correct that the short-term (now conditional) case does not account for probability of eruption, and that the probabilities are used to weight the outputs for the calculation of absolute (previously long-term) exposure. The inclusion of 'conditional' on original line 331 was confusing. We have removed the word conditional and tried to make the text clearer and more explicit in this section.

_Revised text:_ The hazard modelling described thus far provides conditional outputs, i.e. they provide the spatial area affected by a given hazard assuming that an eruption of a given VEI or flow volume has occurred. This is valuable information for crisis planning in the event of unrest; however, comparing across volcanoes at the regional scale requires estimating exposure as a function of the eruption frequency, or probability of occurrence. We achieved this by following the methodology of Hayes et al. (in prep), which uses a Bayesian update and model combination framework to estimate the annual probability of each VEI for each volcano, and the uncertainty around that value (VEI annual probabilities at the 10%, 50% and 90% probability are provided for each volcano in Supplementary Material SM2). Analogue annual eruption probabilities were first calculated using two volcano analogue classification systems (Whelley et al. 2015; Jenkins et al. 2018). These probabilities are then updated separately using the volcano-specific eruption record sourced from GVP (version 4.8.5, downloaded 20 January 2020). This produces two separate frequency-magnitude probability distributions for each volcano, based on the two analogue systems and incorporating volcano-specific eruption data. These two probability distributions are then combined using a model stacking approach to produce a single frequency-magnitude probability distribution for each volcano, with uncertainty. The 50% annual probabilities for each VEI were used in our study to weight the exposure calculated for each VEI scenario, i.e. each exposure value was multiplied by the annual probability of an eruption of that VEI at that volcano occurring, with the sum across them providing the absolute exposure value, which represents the averaged annual exposure across all eruption simulations and scenarios for that volcano and hazard. Incorporating eruption frequency allowed us to better assess the exposure over given timescales, for example multiplying the absolute exposure values by 100 gives the averaged exposure over a 100-year timeframe. For dome collapse PDCs, where flow volume cannot be linked to VEI, we do not incorporate eruption frequency, only providing conditional probabilities.

- **In this light, also lines 603-611 could improve in clarity: in fact, it is not a matter of annual probabilities of VEI3, 4 and 5 separately that change the relative ranking of Cereme and Raung: it is the fact that an explosive eruption from Raung is much more likely (any size, it is the sum of the three annual probabilities that is much larger for Raung than Cereme, so I think that the lines 608-611 could improve in clarity if the sum of the three VEIs' probability are mentioned, rather than the three separate).**

We agree that the probability of an explosive eruption at Raung is greater than at Cereme and this is a major driver behind the order of magnitude difference in the annual probabilities for each individual VEI. However, since we report specific ranking outcomes for individual VEI, we consider it important to mention each annual probability of VEI separately in this paragraph. To

address the reviewer's concern, we have added some text to the paragraph that we hope helps clarify this point:

*Additional text:* Raung is a more frequently erupting volcano, with 62 recorded eruptions since 1800, compared to just three at Cereme in the same time period. This translates into a higher annual probability of an explosive eruption for each VEI at Raung than Cereme, which considerably influences the exposure rankings for these two volcanoes.

**4. Where official hazard maps are available (lines 103-105), I wonder whether it would be possible to compare them with the hazard output from the present study. I think this would strengthen the approach.**

This is something we had discussed in detail during the analysis of our results, and we collected all available hazard maps for our study volcanoes to aid our thinking on it. We came to the conclusion that comparison between our results and officially derived hazard maps was not appropriate for three main reasons: i) we use different methodologies: our approach is probabilistic, while the vast majority of the official hazard maps for our study volcanoes are deterministic; ii) Hazard maps use different, typically volcano-specific data (e.g. geological) as their basis and have a different timeframe of reference; and iii) comparison of our probabilistic outputs with official hazard maps could imply that there is a 'correct' map and that one is being calibrated or validated by the other. In reality the purpose of a hazard map or our maps, and the expected end-users of each, are very different so that we would not expect the maps to be the same. We are therefore not comfortable comparing the two maps publicly although we do find strong similarities between the two that reinforce the importance of major hazard features like topography.

As the comparison of hazard maps with our outputs is something raised by the reviewer and something that we had independently thought of, it is likely something that other readers will also question. We have therefore included some discussion around why we feel comparison is not appropriate in Section 3.1: Case studies:

*Additional text:* We do not compare our maps with official CVGHM or PHIVOLCS hazard maps where they are available for our study volcanoes for the following reasons: i) Comparison implies that one can be calibrated or validated by the other; but ii) We use different methodologies (probabilistic vs. deterministic); iii) Our input data (i.e. analytical vs. geological) are different; and iv) The purpose and expected end-user is not the same.

**5. Finally, somewhere in the Discussion it should be acknowledged that the use of fixed ESPs as representative for a spectrum of similar-scale scenarios is justified at proximal-medium distances, where it tends to produce a hazard assessment that is conservatively higher than when considering a continuous spectrum of ESPs; however we must be aware that it can significantly underestimate hazard assessment in the distal areas, where the impact of the lower-probability upper-end members (that is, within the same VEI class of the representative scenario considered with specific ESPs, but at the upper end of the VEI class and not in the middle) are predominant (e.g., Sandri et al, 2016, Nat Sci Rep).**

We consider the use of single ESPs a limitation of a regional approach and something that should (and typically is) be used in a single volcano assessment. Thus, we included the following bulletpoint in Section 5.1: Hazard approach limitations:

*Additional text:* Simulating with a continuous spectrum of, rather than fixed, ESPs for each VEI scenario. This is particularly important for capturing the larger exposure estimates, as ESPs that

represent the upper end of a VEI, while lower probability than the fixed ESPs we chose or those at the lower end of a VEI, are more likely to produce the larger footprints and thus the larger exposure values (Sandri et al., 2016).

**Minor points:**

- **Line 38: "Of these Southeast" --> "Of these recorded Southeast"**

Corrected

- **Line 39: "in reflecting previous eruptive activity": I am not an English native speaker, but it seems to me that this sentence is not very clear. I suggest changing it into "in reflecting our knowledge of the previous eruptive activity" or something like this.**

Changed to something very similar: "Of these recorded Southeast Asian eruptions, 93% (n=1,435) have occurred since 1500 CE, showing the dominance of historical records reflected in our knowledge of the previous eruptive activity."

- **Table 1: shouldn't the two companion papers by Magill & Blong (2005) on Bull Volcanol be added here? Or you consider them as a "studies that considered the hazard to a site, rather than from a volcano" (as mentioned in the caption)? (not sure if the caption applies to a large city as well)**

Correct, this would be considered a site-specific study. We have made this clearer by adding "such as a city or key infrastructure site" to the Table caption and including Magill and Blong (2005a,b) within our examples.

- **Line 80 and others: I could not find the paper by Biass et al (2014) in the reference list. Please check it all (I did not check all the references one by one)**

Apologies, we have now included Biass et al. (2014) in the reference list and checked all references, resulting in some additional corrections to formatting and inclusion.

- **Line 143: "differing probabilities" --> "differing exceedance probabilities"**

Done.

- **Section 2.1: to me it appears super technical. I suggest moving this section to Appendix.**

We removed the technical components within this section as they are already on the documentation page for the code: https://vharg.github.io/VolcGIS/; however we kept the details regarding the concept and workflow behind the GIS framework as we felt it was critical to understanding how the different information was combined.

*Revised text:* We have developed a geospatial python framework that can source multiple derived hazard and exposure datasets of varying resolution, unifying them to one consistent grid (Figure 2). For each volcano, the extent of the study area was defined based on the bounds of the 1 kg/m$^2$ tephra isomass occurring at a 10% probability for a VEI 5 eruption (see Section 3.2.1). The GIS first applies preprocessing functions to both hazard model outputs and exposure datasets to i) ensure that input files are projected onto the same WGS84 UTM zone as the target volcano, ii) depending on geographic extent of the input file, either crop it to the extent of the study area or pad it with noData value, and iii) resample the input file to a specified spatial resolution. This preprocessing step produces a set of files with consistent geographic references (i.e. coordinate system, extent and pixel resolution) and equal numbers of pixels in x and y directions. This step is critical to ensure that the spatial index of pixels is consistent amongst all files, after which exposure is estimated by translating each pixel's spatial index of hazard footprints onto exposure datasets. Resampling of the rasters is achieved using

a cubic interpolation for continuous hazard data and a nearest neighbor interpolation for discrete exposure data. After resampling, population data are multiplied by the square of the ratio between original and final resolutions in order to scale population counts to the new pixel surface area. Here, a pixel size of 90 m was adopted to keep computing and storage requirements reasonable while retaining a high enough resolution to allow detailed analysis of exposure. The source code of the GIS framework is available at github.com/vharg/VolcGIS.

- **Line 187: again, I cannot refer to Biass et al (2014) with certainty as it is not in the ref list, but I think here it would be fair to mention Folch et al, 2009, Comput Geosci**

The reference for Biass et al. (2014) is used above Folch et al. (2009) as Biass et al. (2014) used Fall3D to carry out a probabilistic assessment, not because it is a kind of 3D model. Many apologies that this was not clear because we omitted the Biass et al. (2014) reference. As it is specific to the probabilistic, and not model type, statement we leave it with this example reference but also include a newly published paper that also carried out probabilistic modelling using Fall3D (Titos et al., 2022, NHESS).

- **Line 194: in this study, aleatoric uncertainty is addressed, but not epistemic (fixed ESPs are assumed, one simulator per hazardous event). So I think it would be correct to remove the word "epistemic" from here.**

We have removed reference to both aleatoric and epistemic sources of uncertainty as there are aspects we are capturing (and missing) from both types.

- **Line 198: I think Appendix A is very important. If allowed (in terms of number of pages and floating bodies of the paper) I would suggest to include it in the main text.**

Due to the large size of Appendix A (3 pages), we offer a compromise and include the key ESPs in a summary table in Section 2.2: Hazard modelling, while keeping the full Appendix A for those who want more detail.

*Additional text and table:* More detail on our modelling approach is provided in the following subsections; we summarise the key ESPs across all hazards in Table 3, with full details and rationale in Appendix A.

**Table 3: Key model input parameters used for the four hazards, with more details and rationale provided in the below subsections and Appendix A.**

| Tephra fall (modelled using Tephra2; Bonadonna et al., 2005) | | | |
|---|---|---|---|
| | **VEI 3** | **VEI 4** | **VEI 5** |
| **Simulations (n)** | 2,880 | 2,880 | 2,880 |
| **Erupted mass (kg)** | $3.2 \times 10^{10}$ | $3.2 \times 10^{11}$ | $3.2 \times 10^{12}$ |
| **Plume height (km)** | 13 | 20 | 27 |
| Large clast (modelled using Rossi et al., 2019) | | | |
| | **VEI 3** | **VEI 4** | **VEI 5** |
| **Simulations (n)** | 2,880 | 2,880 | 2,880 |
| **Plume height (km)** | 13 | 20 | 27 |
| **Clast density (g/cm$^3$)** | 2.5 | | |
| **Clast diameter (cm)** | 3 | | |
| Dome collapse PDC (modelled using LAHARZ modified for use with PDCs (Schilling, 1998; Widiwijayanti et al., 2009)) | | | |
| | **Small volume** | | **Large volume** |
| **Simulations (n)** | 72 | | 72 |
| **Volume (m$^3$)** | $4.5 \times 10^5$ | | $9.8 \times 10^6$ |

| Column collapse PDC (modelled using ECMapProb; Aravena et al., 2020) | | | |
|---|---|---|---|
| | VEI 3 | VEI 4 | VEI 5 |
| Simulations (n) | 300 | 300 | 300 |
| Column collapse height (m) | 1300 | 2000 | 2700 |
| H/L ratio | 0.24 | | |

- **Line 214-215. The sentence "The ESPs ... across volcanoes." does not make much sense. I am not sure in what way fixed ESps ensure convergence. In any case, results across volcanoes would be comparable even if ESPs would not be fixed but sampled from distributions, and still would be if these pdfs would be different and tuned on each volcano according to its eruptive behaviour. I suggest removing this sentence.**

Removed.

- **Line 255: this is important: it is not clear to me how the authors link the two values for the initial flow volume to the annual rates of VEI3, 4 and 5... Do they assign them a relative likelihood to the two volumes, and this is fixed regardless the VEI class? This should be clarified, at least I did not find it in the text.**

We cannot link them as we found no clear relationship between flow volume and the VEI of the associated eruption. Therefore, we provided no absolute probabilities for dome collapse PDC - hence no 'A' column (ranking using absolute probability) in Figure 14. We have added some text to section 2.3 (Incorporating eruption frequency) to make that clear.

*Revised text:* The hazard modelling described thus far provides conditional outputs, i.e. they provide the spatial area affected by a given hazard assuming that an eruption of a given VEI or flow volume has occurred. ... For dome collapse PDCs, where flow volume cannot be linked to VEI, we do not incorporate eruption frequency, only providing conditional probabilities.

- **Line 340 and 350: it could help the reader to know how these numbers (15240 hazard outputs, 11400 exposure estimates, then 498 probabilistic hazard outputs) come out. I can guess 3 VEIs x 3 hazards x 40 volcanoes + 2 volumes x 1 hazard x 40 volcanoes, for the probabilistic hazard outputs, but it does not match...**

We have now included the breakdown for the final numbers in the text, as per the example above, and corrected the erroneous value of 498 to 381 (15240/40 volcanoes).

*Revised text:* The multi-hazard and multi-exposure analysis presented here required nearly 700,000 individual simulations and produced 26,640 probabilistic outputs, comprised of:

*15,240 hazard estimates*: 40 volcanoes x 3 probabilities x [(3 VEI scenarios for column collapse PDC) + (2 flow volumes x 2 buffers for dome collapse PDC) + (3 VEI scenarios for large clast) + (3 VEI scenarios x 3 thickness thresholds x (12 individual months + 1 whole year average wind conditions))];

*11,400 exposure estimates*: 5 exposures x 40 volcanoes x 3 probabilities x [(3 VEI scenarios for column collapse PDC) + (2 flow volumes x 2 buffers for dome collapse PDC) + (3 VEI scenarios for large clast) + (3 VEI scenarios x 3 thickness thresholds x 1 whole year average wind conditions)].

- **Line 356: "Our modelling finds..." --> "Our modelling confirms..."**

Revised.

- **Figure 4a: what are the two grey lobes (one to the west and one to the north)? The buffer? Also, I cannot find in the panel the 2010 PDCs mentioned in Lines 365-367.**

The light grey lobes to the south-southeast, north and northwest represent low probability PDC inundation paths, i.e. the white on the colour bar on the left. The translucency of the lobes was transmitted in the legend, and we have now revised our legend to make it easier to match with the figure itself.

With regards the 2010 PDC footprints, we felt that putting them on the figure would make it too busy and mean that the main information we discuss, i.e. in the more proximal area, would become too hard to see (as the 2010 PDCs went about 4 km off the bottom of the map). We therefore included "(not shown)" in the text to make it clear we don't include the 2010 PDC footprints on the map. We don't believe that omitting the 2010 footprints is detrimental to Figure 4a as the main point of discussing them in the text is that they had unusually long runouts, and the distances are given in the text.

- **Lines382-384: "Figures 4e and 4f show two different ways of plotting our tephra fall results: i) as the probability of exceeding a certain tephra load (≥5 kg/m 2 in Figure 4e), and ii) as the tephra load expected at a given probability (50% in Figure 4f)" --> "Figures 4e and 4f show: i) the probability of exceeding a certain tephra load (≥5 kg/m 2 in Figure 4e), and ii) the tephra load expected at a given probability (50% in Figure 4f)"**

Revised.

- **Line 389: "low probability scenario" --> "low exceedance probability"**

Corrected.

- **Line 403: " For all regions and all hazards, the distribution of population exposure is often asymmetrical" --> "For all regions and all hazards, the distribution of population exposure across different volcanoes is often asymmetrical"**

Revised.

- **Line 413" "than for tephra fall." --> "than for tephra fall, as expected."**

Revised.

- **Line 417: New line after "volcanic islands."**

Revised.

- **Lines 419 and 421: I think Figure 8 is referred for the first time after Figure 9.**

Figures 6,7,8 are referred to altogether in section 3.2 (original line 401), before we refer to Figure 9 on original line 403, before both references on lines 419 and 421. This numbering (Figures 6,7,8 and then 9 around original lines 401-403) is preserved with our revised manuscript, so we think this is technically ok and have not changed it.

- **Captions of Figures 5, 6, 7, 8, 9 (and maybe elsewhere): these are not "conditional probability of occurrence of 50%" but "conditional exceedance probability of 50%", I think. Please change if it is so.**

Correct. Changed, thank-you. We also changed the text to include exceedance probability in the Figure 4 caption.

- **Figures 6 to 9: these plots are very "dense". To improve readability, could the authors specify whether darker bars are "covered" by lighter bars? I mean, do the bars overlap, correct? They do not represent a relative proportion, right? These bar plots may be confusing, as sometimes they are used for relative frequencies, but I guess this is not the case. I think it should be made clear to improve readability.**

Agreed! Plotting the large amount of data in the one place was really challenging and dense bar charts was our best solution. Correct that they overlap and are not stacked. We have revised the text in figure captions 6 through 9 as below (example for Figure 6):

_Revised text:_ Exposure to tephra fall accumulation summarised per region and exposure type for a conditional exceedance probability of occurrence of 50%. Overlying (not stacked) bars illustrate the variability of exposure with VEI (with the top of the bar representing exposure for that VEI) and dotted lines the median for the region. Note that specific thresholds of tephra loads (as defined in section 2.2.1) are used for various exposure types.

**Line 492: I think here it would be fair to add Macedonio & Costa, 2021, NHESS**

Macedonio and Costa, 2012 added.

- **Line 509: again, 50% exceedance probability**

Corrected.

- **Lines 508-513: Actually, in Java panel in the top part of Figure 10, Krakatau seems to be the one with the smallest (close to 0) monthly variation in the population exposure to tephra fallout (VEI 4, P=50%). I do not see the large variation in January at all at Krakatau. I see it for Guntur or Papandayan, for example. Please clarify.**

The figure showed absolute values so the change in population exposure for Krakatau, while being large in the relative sense (~23-fold increase), is small in absolute values (0.27 x $10^6$ in January), especially when compared to the volcanoes with much larger exposure, meaning it was difficult to see in the figure. We have resolved this by including a second plot within Figure 10 to show the relative variability. This new panel highlights the large change for Krakatau - and that the rest of the volcanoes have relatively smaller (but still large: up to 600%) variability. We have included more text within Section 3.2: Hazard seasonality to discuss this.

_Revised text:_ Across all study volcanoes, relative changes to population exposure estimates as a result of seasonal variability are typically positive, and within 150% of the whole year estimate across Indonesia (with the notable exception of Krakatau) but up to 600% in the Philippines, with Camiguin and Pinatubo showing the largest relative changes (Figure 10).

Three volcanoes in the region best illustrate changes in population exposure as a function of the month of the eruption. Firstly, an eruption at Krakatau volcano in January leads to a relatively drastic increase in population exposure compared to the rest of the year. Considering a VEI 4 eruption and a 50% exceedance probability of occurrence, an eruption in January leaves ~270,000 people exposed to an accumulation ≥1 kg/m² compared to ~11,000 (a 2,350% increase) when all months are aggregated. Population exposure throughout the year at Krakatau is typically low relative to the other volcanoes in our study as winds predominantly disperse tephra towards the west and over the sea, hence the small absolute variability. Wind conditions below ~15 km in January blow mostly to the north or west-southwest, reducing the westward extent of the ≥1 kg/m² isopach and extending it eastwards, affecting human settlements on the western parts of Java. A similar behaviour is observed at Guntur volcano; although the dominant wind direction is towards the southwest, winds during the rainy months (Dec-Apr) also display dispersal towards

the north and the east, which increase the probability of Bandung (9 million people, northwest of Guntur) and Garut (100,000 people, southeast of Guntur) being affected by ≥1 kg/m² of tephra, leading to a relatively small, but absolute large increase in exposure. Finally, winds at Taal volcano show a strong northward component around the tropopause (∼8 to 15 km) during the peak dry season (e.g. January) compared to the rest of the year, when winds at this height mostly blow towards the west. As a result, eruptions during the month of January increase the probability of tephra deposits affecting Metro Manila, as demonstrated by Taal's January 2020 eruption.

*Revised Figure 10:*

[Figure]

Monthly variability in population exposure to tephra accumulations ≥ 1 kg/m² (VEI=4, P=50%)

- **Figure 11: In panel for VEI5 and tephra fall, there are names of volcanoes that are not referred to in the caption. I guess they are the volcanoes scoring the largest value for that combination, i.e., the ones giving the largest grey circle. However, it should be mentioned in the caption, if so. And the names should be given in all panels (I guess they change from case to case).**

Apologies, the volcano names were left on the plot in error, we originally investigated whether this figure could include the names of the top ranked volcano for that category (e.g. Cereme has the highest ranked tephra fall hazard at VEI 5) but we felt this would make the plots too busy as we'd need names on each panels. We have therefore removed the names from panel 1. Thanks for catching that.

- **Lines 670-671: "With this study we have evaluated multiple categories of exposure (n=5) to a range of volcanic hazards (n=4) and VEI scenarios (n=3) to give probabilistic**

**outputs for 40 high-threat volcanoes" --> "With this study we have evaluated five multiple categories of exposure to a range of four volcanic hazards and three VEI scenarios to give probabilistic outputs for 40 high-threat volcanoes."**

Revised to something similar:

*Revised text:* With this study we have evaluated five categories of exposure to four volcanic hazards and three VEI scenarios to give probabilistic outputs for 40 high-threat volcanoes.

- **Line 673: The "long-term" or "absolute" assessment does not "accounts for the probability of occurrence of a given eruption scenario at each volcano", but for the probability of occurrence of the different eruption scenarios considered, at each volcano.**

Corrected.

- **Line 708: in the list of areas to widen the assessment, I would reiterate the extension to other relevant hazardous events, such as lahars and tsunamigenic eruptions.**

Done.

[revised manuscript text omitted]